# WIPI3 and WIPI4 β-propellers are scaffolds for LKB1-AMPK-TSC signalling circuits in the control of autophagy

Daniela Bakula[1,2,*], Amelie J. Müller[1,2,*], Theresia Zuleger[1], Zsuzsanna Takacs[1,2], Mirita Franz-Wachtel[3], Ann-Katrin Thost[1], Daniel Brigger[4], Mario P. Tschan[4], Tancred Frickey[5], Horst Robenek[6], Boris Macek[2,3] & Tassula Proikas-Cezanne[1,2]

Autophagy is controlled by AMPK and mTOR, both of which associate with ULK1 and control the production of phosphatidylinositol 3-phosphate (PtdIns3P), a prerequisite for autophagosome formation. Here we report that WIPI3 and WIPI4 scaffold the signal control of autophagy upstream of PtdIns3P production and have a role in the PtdIns3P effector function of WIPI1-WIPI2 at nascent autophagosomes. In response to LKB1-mediated AMPK stimulation, WIPI4-ATG2 is released from a WIPI4-ATG2/AMPK-ULK1 complex and translocates to nascent autophagosomes, controlling their size, to which WIPI3, in complex with FIP200, also contributes. Upstream, WIPI3 associates with AMPK-activated TSC complex at lysosomes, regulating mTOR. Our WIPI interactome analysis reveals the scaffold functions of WIPI proteins interconnecting autophagy signal control and autophagosome formation. Our functional kinase screen uncovers a novel regulatory link between LKB1-mediated AMPK stimulation that produces a direct signal via WIPI4, and we show that the AMPK-related kinases NUAK2 and BRSK2 regulate autophagy through WIPI4.

[1] Department of Molecular Biology, Interfaculty Institute of Cell Biology, Eberhard Karls University Tuebingen, D-72076 Tuebingen, Germany. [2] International Max Planck Research School 'From Molecules to Organisms', Max Planck Institute for Developmental Biology and Eberhard Karls University Tuebingen, D-72076 Tuebingen, Germany. [3] Proteome Center Tuebingen, Interfaculty Institute of Cell Biology, Eberhard Karls University Tuebingen, D-72076 Tuebingen, Germany. [4] Division of Experimental Pathology, Institute of Pathology, University of Bern, CH-3008 Bern, Switzerland. [5] Department of Biology, Applied Bioinformatics, Konstanz University, D-78457 Konstanz, Germany. [6] Institute of Experimental Musculoskeletal Medicine, University Hospital Muenster, D-48149 Muenster, Germany. * These authors contributed equally to this work. Correspondence and requests for materials should be addressed to T.P.-C. (email: tassula.proikas-cezanne@uni-tuebingen.de).

Autophagy[1–4] is regulated by AMPK and TORC1, which activate catabolic and anabolic pathways, respectively, and interact to control metabolism and maintain energy homeostasis[5,6]. In the presence of amino acids and growth factors, TORC1 becomes activated at the lysosomal surface[7]. Activated TORC1 inhibits autophagy through the site-specific phosphorylation of the autophagy initiator protein kinase ULK1 (refs 8,9). TORC1-mediated autophagy inhibition is released in the absence of amino acids and is achieved through the action of the TORC1 inhibitor complex TSC1–TSC2 (refs 10–12), which provokes the displacement of TORC1 from lysosomes[13,14]. TSC complex activation is regulated through LKB1-mediated AMPK activation[15], which phosphorylates TSC2 (ref. 16). In addition, AMPK activates ULK1 through direct phosphorylation[9,17], and in turn, ULK1 phosphorylates components of the phosphoinositide-3 kinase class III (PI3KC3) complex[18,19], allowing phosphatidylinositol 3-phoshpate (PtdIns3P) production, a prerequisite for autophagosome formation[19–23].

In humans, the PtdIns3P effector function in autophagy is attributed to the four WIPI proteins, representing the human group of proteins within the PROPPIN protein family[24]. WIPI2 functions as a PtdIns3P effector[25,26], bridging PtdIns3P production with the recruitment[26] of the ATG16L[27,28] complex for LC3 (refs 29,30) lipidation and subsequent autophagosome formation[25,26,31]. WIPI1 (ref. 32) is considered to function upstream[33] and WIPI4 downstream of LC3 (ref. 34); however, their functions are unknown, and WIPI3 is uncharacterized[24].

Despite the notion that glucose starvation induces autophagy through AMPK-mediated ULK1 phosphorylation[9,17], which acts upstream of WIPI1 and WIPI2 (refs 26,35), neither WIPI1 nor WIPI2 respond to glucose starvation[36,37]. Here, we demonstrate that glucose starvation signals via the LKB1-AMPK network to WIPI4 in complex with ATG2, which in response contributes to the regulation of autophagosome formation. WIPI3 is also under the control of AMPK as it associates with activated TSC complex in controlling mTOR activity in the lysosomal compartment. Hence, both WIPI4 and WIPI3 function upstream of PtdIns3P production but also downstream of WIPI1-WIPI2 in controlling the size of nascent autophagosomes, with WIPI4 acting in association with ATG2 and WIPI3 in association with FIP200. Our study, a combined protein interactome and kinome screening approach, reveals that the four human WIPI proteins function as a scaffold circuit, interconnecting autophagy signal control with autophagosome formation.

## Results

**WIPI3 and WIPI4 bind PtdIns3P at nascent autophagosomes.** With regard to the reported features of WIPI1 and WIPI2 (refs 25,26,32,38), we assessed WIPI3 and WIPI4 by comparative[39,40] structural modelling (Fig. 1a), phospholipid-protein overlay assessments[38] (Fig. 1b) and subcellular localization using fluorescence-based confocal laser-scanning microscopy (LSM) (Fig. 1b–f). Structural homology modelling using HHpred[41] revealed that all WIPI members fold into seven-bladed β-propellers with an open Velcro topology[32] (Fig. 1a). Of note, we used a new WIPI3 sequence in the current study, as our original WIPI3 cloning isolate[32] proved to represent an N-terminal-truncated version (see Supplementary Note, Supplementary Fig. 1a–c,h).

As reported[42], the binding of WIPI1 and WIPI2 to PtdIns3P at nascent autophagosomes is demonstrated by the appearance of subcellular fluorescent puncta (Fig. 1b, right panels, Supplementary Movies 1 and 2; Supplementary Fig. 1e). The number of cells displaying GFP-WIPI1 (refs 32,38) and GFP-WIPI2B[31,32] puncta significantly increased upon starvation and decreased upon PI3K inhibition (Fig. 1c), and GFP-WIPI1 and GFP-WIPI2B puncta co-localized with myc-ATG14 (refs 43,44), myc-DFCP1 (refs 31,45), ATG12 (refs 46,47), LC3 (ref. 48) and p62 (ref. 49), as expected (Fig. 1d, Supplementary Fig. 1f). The numbers of GFP-WIPI1- and GFP-WIPI2B-puncta-positive cells further increased in the presence of the lysosomal inhibitor bafilomycin A1 (ref. 50) (Fig. 1c), in line with a previous report on the localization of WIPI1 and WIPI2 at autophagosomes[51].

GFP-WIPI3 and GFP-WIPI4 puncta were smaller in size and less complex than GFP-WIPI1 and GFP-WIPI2B puncta (Fig. 1b, right panels, Supplementary Movies 3 and 4; Supplementary Fig. 1e). The number of GFP-WIPI3-puncta-positive cells significantly increased upon starvation and bafilomycin A1 administration and decreased upon PI3K inhibition (Fig. 1c). The appearance of endogenous WIPI3 puncta upon starvation was also apparent (Supplementary Fig. 1g). In the following, we employed GFP-tagged WIPI3, as proper folding was confirmed through specific binding to PtdIns3P and PtdIns(3,5)P$_2$ (Fig. 1b, Supplementary Fig. 1b–d).

The number of GFP-WIPI4-puncta-positive cells increased upon starvation (Fig. 1c), but this increase was not abolished via PI3K inhibition (Fig. 1c; Supplementary Fig. 1i). Endogenous WIPI4 puncta were already prominent in fed conditions, and they were significantly increased upon starvation (Supplementary Fig. 1j). Both GFP-WIPI3 and GFP-WIPI4 were clearly localized at nascent autophagosomes, as indicated by their co-localization with the markers used (myc-ATG14, myc-DFCP1, ATG12, LC3 and p62) (Fig. 1d, Supplementary Fig. 1f). However, the co-localization of GFP-WIPI4 with myc-ATG14, myc-DFCP1 and endogenous ATG12 was less prominent than that of other WIPI members (Fig. 1d, Supplementary Fig. 1f). Based on these findings, all WIPI proteins appear to co-localize at nascent autophagosomes. Indeed, the co-localization of endogenous WIPI2, WIPI4 and GFP-WIPI1 (Fig. 1e) and GFP-WIPI3 with endogenous WIPI4, myc-WIPI1 and myc-WIPI2B (Fig. 1f) was apparent.

In summary, all four WIPI members fold into seven-bladed β-propeller proteins that bind PtdIns3P and localize at nascent autophagosomes upon starvation.

**Autophagosome formation requires WIPI3 and WIPI4.** Using the human melanoma cell line G361, which expresses detectable levels of all endogenous WIPI members, we generated stable WIPI knockdown (KD) cell lines (Fig. 2a, Supplementary Fig. 2a) and scored cells for autophagosomes by electron microscopy (EM) (Fig. 2b, Supplementary Fig. 2b–f, Supplementary Data 1). Compared with the control cell line (shControl, Fig. 2b, Supplementary Fig. 2b), WIPI1 KD cells did produce autophagosomes upon starvation (Fig. 2b, panel shWIPI1; Supplementary Fig. 2c), although we observed a 2.65-fold reduced presence of autophagosomal structures (Supplementary Data 1). In line with this finding, autophagic flux[50] assessments by LC3 lipidation analysis was previously reported to be negatively affected in WIPI1 KD cells[24]. Additionally, we here found that starvation-induced degradation of long-lived proteins[52] was significantly reduced in WIPI1 KD cells (Supplementary Fig. 2g). EM of WIPI2 KD revealed that proper autophagosome formation was negatively affected as inferred from a 4.93-fold accumulation of rough endoplasmic reticulum tubular structures (Fig. 2b, panel shWIPI2; Supplementary Fig. 2d), prominently marking an early blockade of autophagosome formation[26] before phagophore formation (template membranes from which autophagosomes emerge[23,53]) (Supplementary Data 1). In line with this finding,

the appearance of autophagosomal structures decreased by 2.94-fold when compared with control cells (Supplementary Data 1). Moreover, in WIPI2 KD cells, the number of GFP-LC3 puncta was significantly reduced (Fig. 2c), corroborating the previous finding that WIPI2 is required for LC3 lipidation[25,26]. The above results confirm that both WIPI1 and WIPI2 function upstream of LC3, as previously suggested[35]. In addition, the results underline the conception that WIPI2 is required for autophagosome formation, whereas WIPI1 is dispensable to a certain degree.

Interestingly, both WIPI3 KD (Fig. 2b, panel shWIPI3; Supplementary Fig. 2e) and WIPI4 KD (Fig. 2b, panel shWIPI4; Supplementary Fig. 2f) resulted in the appearance of cup-shaped double-membrane structures resembling elongated phagophore formation sites (Fig. 2b, Supplementary Fig. 2e,f; Supplementary Data 1). In both, WIPI3 KD and WIPI4 KD cells the formation of autophagosomal structures decreased, respectively, by 6.42- and 3.47-fold when compared with shControl cells (Supplementary Data 1). In addition, rough endoplasmic reticulum tubular structures accumulated in the absence of WIPI4, as was observed

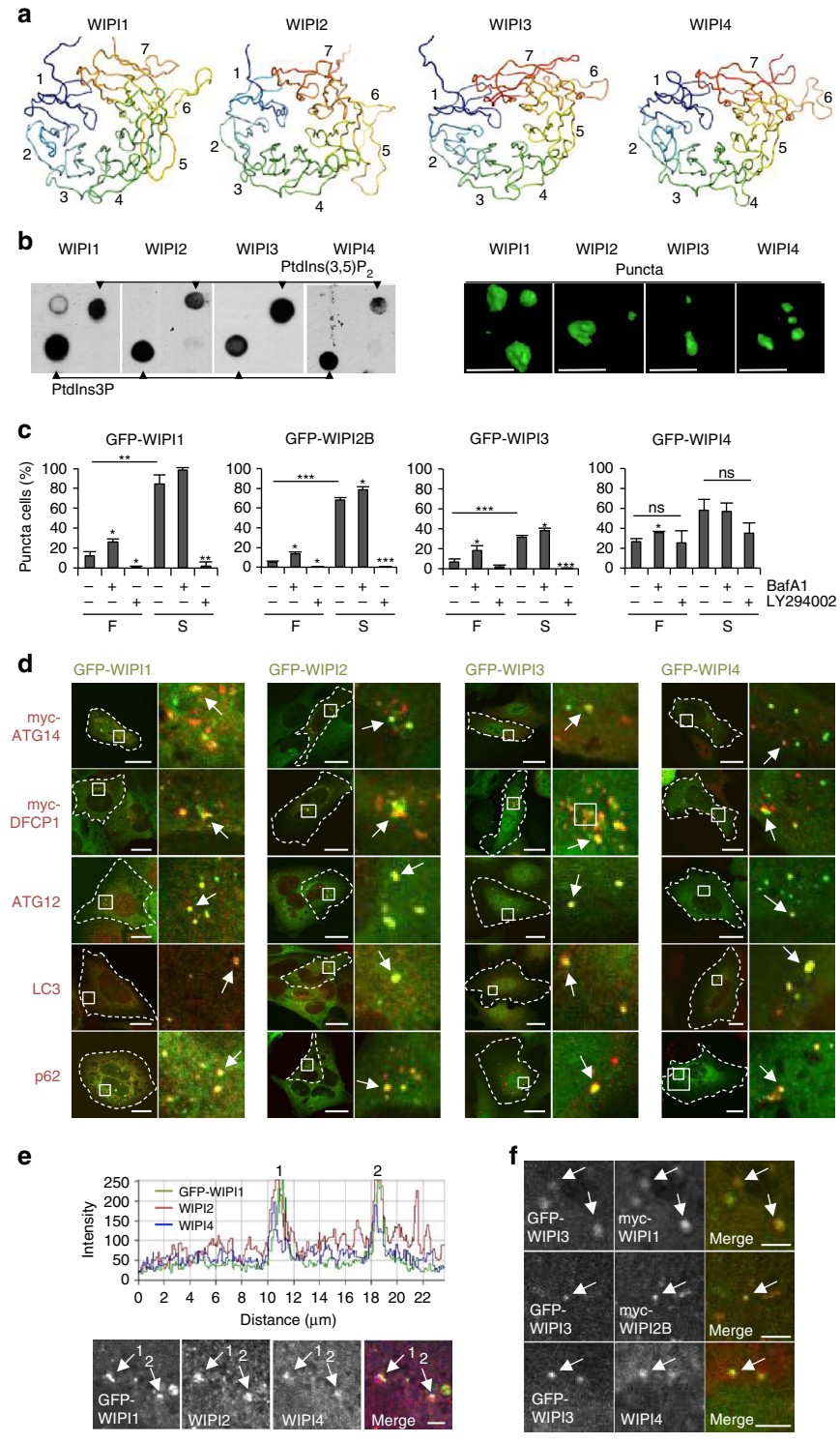

in WIPI2 KD cells (Supplementary Fig. 2f, Supplementary Data 1). These data suggest that both WIPI3 KD and WIPI4 KD blocked autophagosome formation downstream of WIPI2. As WIPI2 recruits the ATG16L complex for LC3 lipidation[26], we anticipated that lipidated LC3 would accumulate in the absence of WIPI3 or WIPI4, which was indeed the case when we looked for GFP-LC3 puncta using automated high-throughput imaging[42] (Fig. 2d). In line with this observation, we also found that endogenous WIPI2 puncta accumulated in WIPI3 KD and WIPI4 KD cells due to the downstream blockade of LC3 (Fig. 2e). Moreover, by western blotting we observed a significant increase in basal levels of lipidated (LC3-II) and unlipidated LC3 (LC3-I) (Fig. 2f, Supplementary Fig. 2h). Upon amino-acid starvation, however, LC3 levels did not significantly change (Supplementary Fig. 2h), indicating a layer of redundancy between the different WIPI proteins.

Of additional note, EM observation of WIPI KD in the G361 melanoma cell line showed that WIPI1 and WIPI2 KD cells were filled with melanosomes that were not detected in control cells (Supplementary Fig. 2i). The appearance of melanosomes upon ULK1 and WIPI1 silencing has been reported[54,55]. Altogether with our observation, this suggests that autophagosome and melanosome formation may be co-regulated[56].

**WIPI β-propellers are distinct protein interaction platforms.** With regard to the co-localization of all WIPI members at nascent autophagosomes upon starvation (Fig. 1e,f), we examined whether WIPI members also associated through protein–protein interactions. Indeed, using GFP- and myc-tagged WIPI variants[31,32], we found that WIPI1, WIPI2B, WIPI2D and WIPI4 were able to associate with each other (Supplementary Fig. 3a,b). Next, we established a WIPI protein interactome using mass spectrometry (MS)[57] upon the immunopurification of GFP-tagged WIPI1–4 proteins (Fig. 3). As a control, we employed GFP alone and excluded the signals from tag-associated proteins from the final results, along with those of proteins giving low peptide counts (see Methods section, Supplementary Data 2). The WIPI protein interactomes were generated using GeneMANIA[58] (Fig. 3a), and interactions identified in this study (red lines) are incorporated with previously reported[59] and predicted interactions (grey lines).

The proteomics analysis demonstrated that all four WIPI proteins associated in a distinct and non-redundant protein network. In addition, WIPI1, WIPI2 and WIPI4 share NudC as a common interaction partner (Fig. 3a,b). We confirmed the specific binding of endogenous NudC[60–62] with WIPI1, WIPI2B

and WIPI4. Consistent with our MS analysis, NudC did not co-purify with WIPI3 (Supplementary Fig. 4).

We concentrated our follow-up investigations on ATG proteins and the known autophagy regulators appearing in our WIPI interactome network (Fig. 3b). In support of our WIPI interactome data, we confirmed the reported interaction between WIPI2 and the ATG16L complex[26] (Figs 3a,b and 4a). Moreover, we found that WIPI2 also interacts with WIPI1 (Fig. 3a,b; Supplementary Fig. 3a,b), and we provide evidence that WIPI1 likely supports WIPI2 in ATG16L1 recruitment for LC3 lipidation at nascent autophagosomes. We show that GFP-WIPI1 associates with endogenous ATG16L (Figs 3a,b and 4a); however, as we found only a few peptide counts for ATG16L in our WIPI1 MS analysis and low amount of ATG16L copurifying with GFP-WIPI1 when compared with GFP-WIPI2B and GFP-WIPI2D (Fig. 4a), we suggest that the interaction between WIPI1 and ATG16L occurs indirectly via WIPI2, in line with previously reported suggestions. Nevertheless, the functional KD of not only WIPI2 (Fig. 4b) but also WIPI1 (Fig. 4c) significantly reduced the number of ATG16L-puncta-positive cells. Moreover, WIPI1 depends on WIPI2 for proper localization to nascent autophagosomes, as demonstrated by the significantly decreased number of GFP-WIPI1 punctate structures in WIPI2 KD cells (Fig. 4d). By contrast, WIPI1 KD had no effect on the number of WIPI2 punctate structures in fed and starved conditions (Supplementary Fig. 3c), suggesting that WIPI2 is required for WIPI1 recruitment and that the presence of the WIPI1-WIPI2 heterodimer may function more efficiently in ATG16L complex recruitment. Moreover, we found that both WIPI1 KD (Fig. 4e) and WIPI2 KD (Fig. 4f) significantly reduced the number of GFP-WIPI4 punctate structures in fed and starved conditions, supporting the idea that WIPI1 and WIPI2 function upstream of WIPI4 during autophagosome formation[24].

In support of a role for WIPI3 and WIPI4 downstream of LC3 (see also Fig. 2d–f), ATG16L puncta accumulated in WIPI3 KD and WIPI4 KD cells (Fig. 4g). Supporting a role for WIPI1-WIPI2 upstream of LC3, both GFP-WIPI1- and GFP-WIPI2B-puncta-positive cells were significantly increased in ATG5 KO conditions due to a blockade of LC3 lipidation (Fig. 4h,i; Supplementary Movies 5–8). This effect was not observed when GFP-WIPI3- and GFP-WIPI4-puncta-positive cells were quantified in ATG5 KO conditions (Fig. 4i).

**WIPI3 associates with the TSC complex and FIP200.** Our WIPI3 protein interactome analysis revealed that WIPI3 likely connects to the autophagic machinery through associations

**Figure 1 | All WIPI members fold into seven-bladed β-propeller proteins that bind PtdIns3P and co-localize at nascent autophagosomes.** (**a**) Structural homology modelling using HHpred. Propeller blades are indicated (1 to 7), and unstructured sequences are omitted. (**b**) Protein-phospholipid binding overlay assays using G361 cell extracts followed by the detection of endogenous WIPI1, WIPI2 or WIPI4 or using a monoclonal U2OS cell line stably expressing GFP-WIPI3 followed by anti-GFP enhanced chemiluminescence (ECL) detection (left panels). Monoclonal U2OS cell lines stably expressing GFP-WIPI1, GFP-WIPI2B or GFP-WIPI3 or transiently expressing GFP-WIPI4 were starved for 3 h with nutrient-free medium. Images were acquired by confocal LSM and processed using Volocity to generate 3D-reconstruction fly-through movies (Supplementary Movies 1–4). Representative still images are presented (right panels). Scale bars: 3 μm. (**c**) U2OS cells stably expressing GFP-WIPI1, GFP-WIPI2B or GFP-WIPI3 or transiently expressing GFP-WIPI4 were fed (F) or starved (S) for 3 h with or without bafilomycin A1 (BafA1) or LY294002 (LY). The percentage GFP-WIPI puncta cells were calculated (up to 429 cells per condition, $n = 3$). (**d**) U2OS cells stably expressing GFP-WIPI1, GFP-WIPI2B or GFP-WIPI3 or transiently expressing GFP-WIPI4 were starved (3 h), and endogenous ATG12, LC3, p62 (anti-ATG12, anti-LC3, anti-p62 and IgG-Alexa Fluor 546 antibodies) or transiently expressed myc-tagged ATG14 or DFCP1 detected (anti-myc/IgG-Alexa Fluor 546 antibodies). Merged confocal LSM images are presented (left panels, dashed lines: cell boundaries; right panels: magnified subsections). Scale bars: 20 μm. (**e**) Confocal LSM examinations of starved (3 h) U2OS cells stably expressing GFP-WIPI1 and immunostained with anti-WIPI2/IgG-Alexa Fluor 546 and anti-WIPI4/IgG-Alexa Fluor 633 antibodies. Intensity profiles of co-localizations (peaks 1 and 2, upper panel) are displayed, along with the corresponding magnified image sections. Scale bar: 2.5 μm. (**f**) U2OS cells stably expressing GFP-WIPI3 and transiently expressing myc-tagged WIPI1 or WIPI2B were starved (3 h) and immunostained using anti-myc or anti-WIPI4 and IgG-Alexa Fluor 546 antibodies. Scale bar: 2.5 μm. Arrows in magnified image sections indicate co-localization events (**d–f**). Supplementary Material is available: Supplementary Fig. 1, Supplementary Note. Statistics and source data can be found in Supplementary Data 1. Mean ± s.d.; heteroscedastic $t$-testing; $P$ values: \*$P < 0.05$, \*\*$P < 0.01$, \*\*\*$P < 0.001$, ns: not significant.

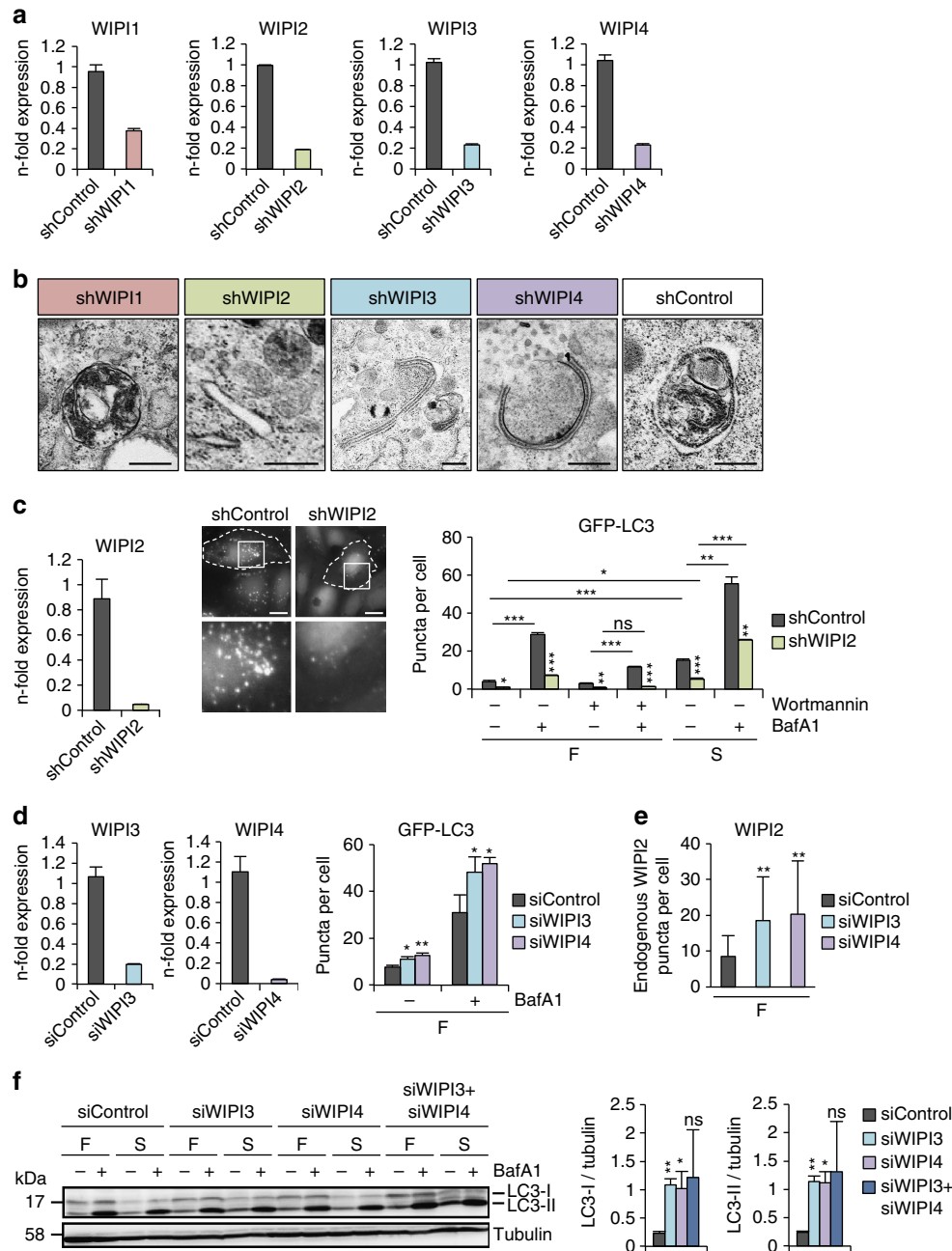

**Figure 2 | Differential contributions of WIPI members to the formation of functional autophagosomes.** G361 cell lines stably expressing shRNAs targeting WIPI1 (shWIPI1), WIPI2 (shWIPI2), WIPI3 (shWIPI3), WIPI4 (shWIPI4) or non-targeting shRNA (shControl) were assessed by quantitative RT–PCR (**a**) or electron microscopy analysis upon starvation (3 h), scale bars: 500 nm (**b**). (**c**) Monoclonal U2OS cell lines stably expressing GFP-LC3 and shWIPI2 or shControl were assessed by quantitative RT–PCR (left panel). Automated high-throughput image acquisition (middle panel, upper row: dashed lines indicate cell boundaries; lower row: magnified sections) and analysis (right panel). The numbers of GFP-LC3 puncta in control (shRNA) or WIPI2-KD (shWIPI2) cells were calculated under fed (F) or starved (S) conditions with or without wortmannin (WM) or bafilomycin A1 (BafA1). The mean number of GFP-LC3 puncta per cell was calculated (up to 15,401 cells per condition, n = 3). Scale bars: 20 μm. (**d**) U2OS cells stably expressing GFP-LC3 were transiently transfected (48 h) with control siRNA (siControl) or siRNAs targeting WIPI3 (siWIPI3) or WIPI4 (siWIPI4). Total RNA was extracted for quantitative RT–PCR (left panel: WIPI3, middle panel: WIPI4). In addition, fed cells (F) cells were treated with or without of BafA1 for automated high-throughput image analysis (right panel). Mean numbers of GFP-LC3 puncta per cell are presented (up to 2,680 cells per condition, n = 3). (**e**) Endogenous WIPI2 puncta formation was examined (anti-WIPI2/IgG-Alexa Fluor 488 antibodies) by confocal LSM and ImageJ (least 15 cells, n = 3) using stable G361 WIPI3-KD (shWIPI3), WIPI4-KD (shWIPI4) and control cells (shControl). (**f**) G361 cells were transiently transfected with control siRNA (siControl), siWIPI3, siWIPI4 or a siWIPI3/siWIPI4 combination and fed (F) or starved (S) for 3 h with or without BafA1. Cell lysates were analysed by anti-LC3 and anti-tubulin western blotting. The migrations of LC3-I and LC3-II are indicated (left panel). LC3-I (middle panel) and LC3-II (right panel) abundances were quantified and results achieved in fed conditions (F) are presented (n = 3). Supplementary Material is available: Supplementary Fig. 2. Statistics and source data can be found in Supplementary Data 1. Mean ± s.d.; heteroscedastic t-testing; P values: $*P < 0.05$, $**P < 0.01$, $***P < 0.001$, ns: not significant.

with the obligatory hetero-dimer TSC1-TSC2 (refs 10,12) and with FIP200 (refs 63–65) (Fig. 3). The specific interaction between WIPI3 and the TSC complex was demonstrated by immunopurification of both endogenous TSC1 and TSC2 with GFP-WIPI3 (Fig. 5a). The findings of this analysis further revealed that GFP-WIPI3 interacts with TSC2 phosphorylated

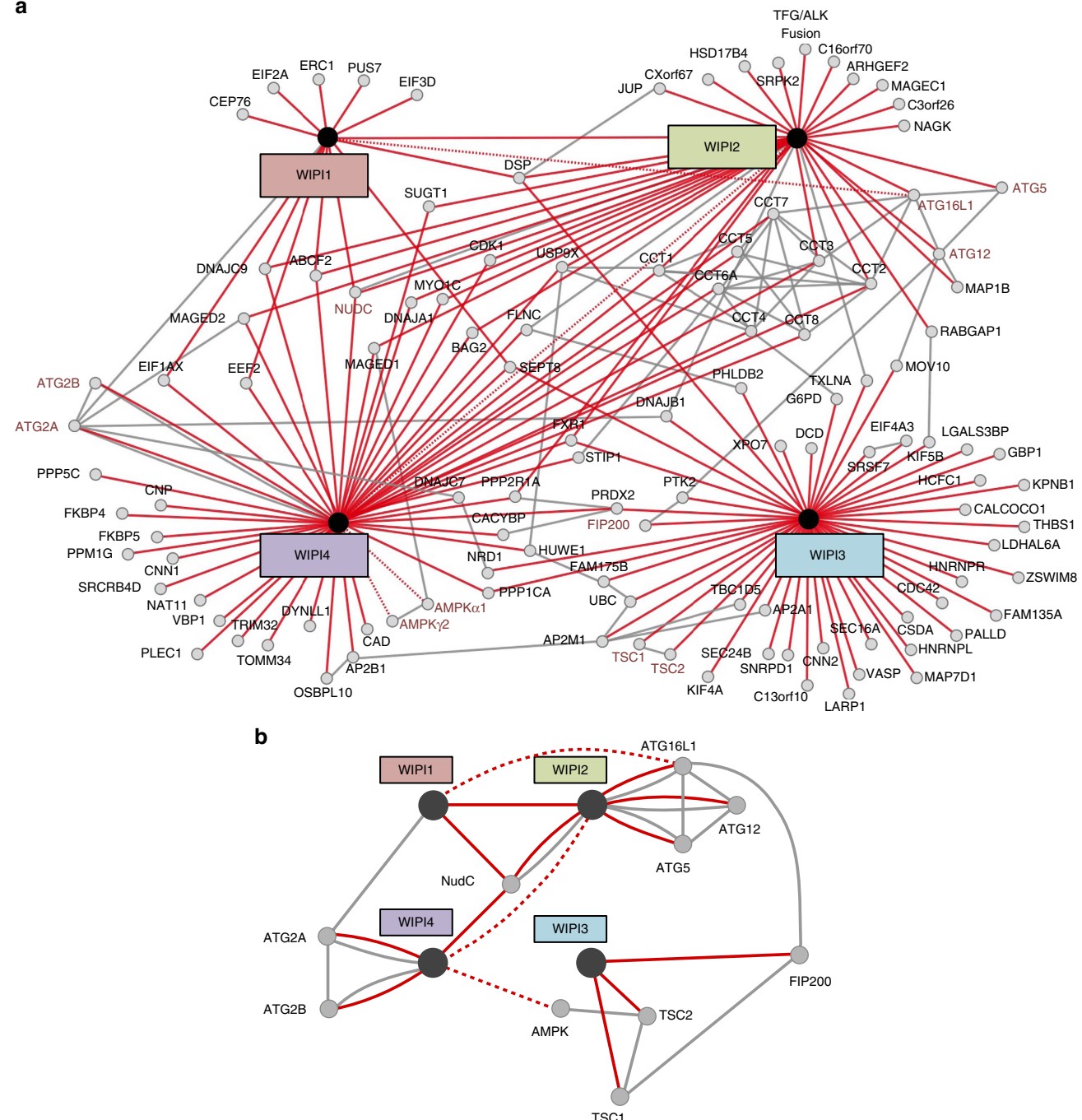

**Figure 3 | The WIPI protein interactome.** Monoclonal U2OS cell lines stably expressing GFP-WIPI1, GFP-WIPI2B, GFP-WIPI2D, GFP-WIPI3, GFP-WIPI4 or GFP alone were analysed by anti-GFP immunoprecipitation followed by nano liquid chromatography (LC)-MS/MS analysis (for detailed information, see Materials and Methods). (**a**) GeneMANIA was used to visualize identified WIPI protein–protein interactions. Red lines represent interactions identified in this study, and grey lines represent previously reported and predicted interactions (GeneMANIA, see Materials and Methods). ATG proteins and autophagy regulators are indicated with red letters. Complete results can be found in Supplementary Data 2. Interactions identified using GFP-WIPI2B and GFP-WIPI2D were combined and are presented as WIPI2 interactions. Proteins exclusively interacting with either of the two WIPI2 isoforms can be found in Supplementary Data 2. (**b**) A sub-network of the WIPI protein interactome focusing on autophagy-related and autophagy-relevant proteins is shown. Red lines represent interactions identified in this study, with dotted red lines representing interactions with low peptide counts in our LC-MS/MS analysis. Grey lines represent previously reported and predicted interactions (GeneMANIA, see Materials and Methods). In addition, the reported interactions of FIP200-ATG16L, TSC1-FIP200 and AMPK-TSC2 that did not appear when using GeneMANIA are also shown in grey. Supplementary Data demonstrating WIPI1, WIPI2B, WIPI2D and WIPI4 co-immunoprecipitation with NudC (Supplementary Fig. 4) is available.

at serine 1,387 (Fig. 5a), an AMPK-specific phosphorylation site required for TSC-mediated TORC1 inhibition[16]. The interaction between the TSC complex and endogenous WIPI3, but not WIPI1, WIPI2 or WIPI4 has also been confirmed (Fig. 5b) and, as anticipated, is independent of ATG5 as shown by using GFP-WIPI3 in mouse embryonic fibroblasts (MEFs) depleted from ATG5 (Fig. 5c). Moreover, we observed that the association of GFP-WIPI3 with the endogenous TSC complex appeared to be more abundant in starved (S) cells (Fig. 5c,d). Furthermore, the association

between GFP-WIPI3 and the TSC complex was not dependent on PtdIns3P production because the complex formed despite the addition of the PI3K inhibitor LY294002 to the starved cells (Fig. 5d). Of note, a greater degree of complex formation was again apparent in starved cells (S, 15′–3 h) than in fed cells (Fig. 5d). Furthermore, we overexpressed full-length TSC1 (TSC-FL), a TSC1 fragment corresponding to the middle part of the protein (TSC1-M), which has previously been shown to interact with FIP200 (ref. 66), or a C-terminal fragment of TSC1 (TSC1-C),

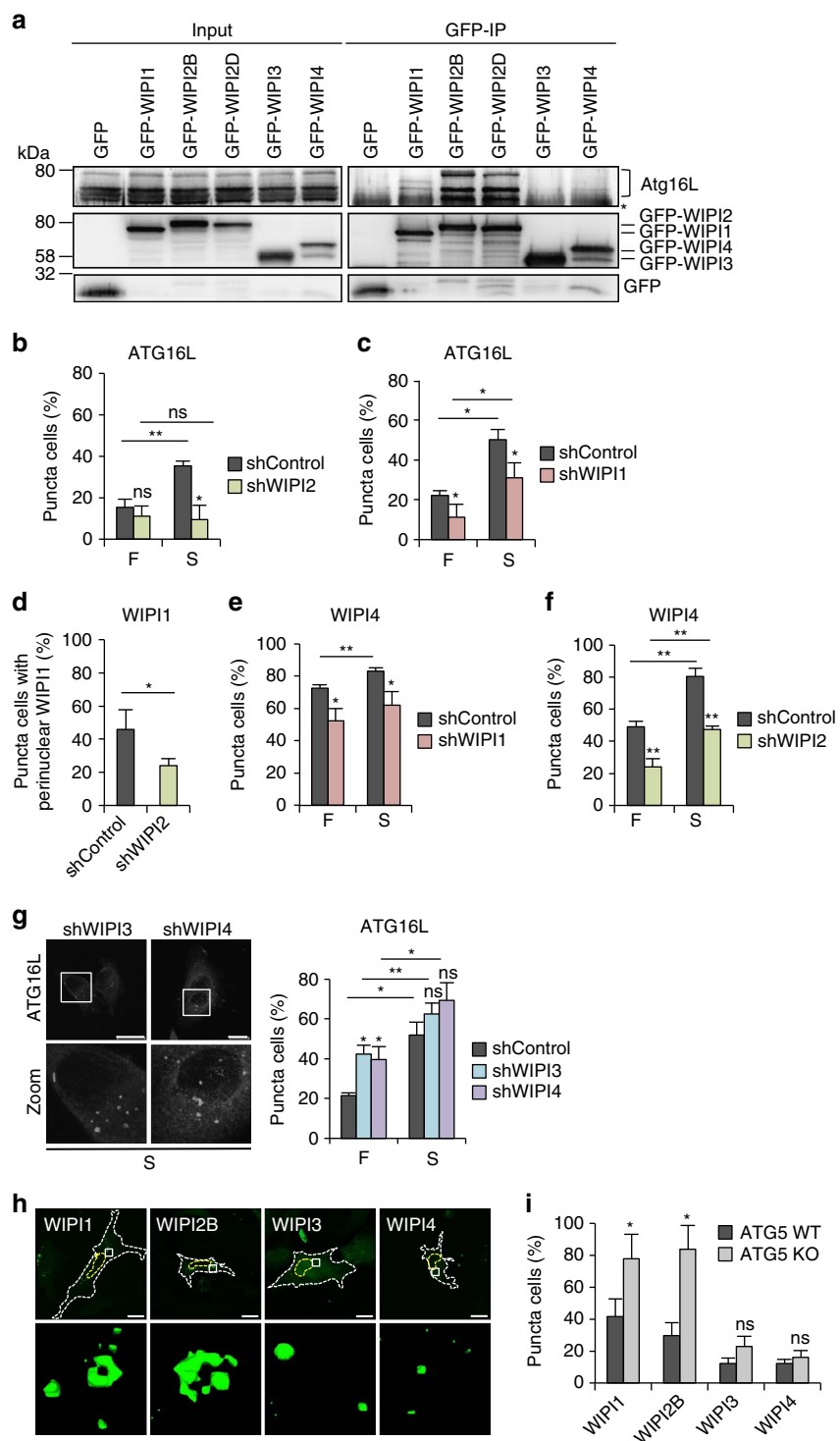

and found that GFP-WIPI3 clearly co-immunoprecipitated with TSC1-M (Fig. 5e).

Because the active TSC complex localizes to lysosomes for TORC1 inhibition, we enriched lysosomes and found that GFP-WIPI3 co-fractionated with endogenous TSC2 and LAMP2 (ref. 67), marking lysosomes/late endosomes (Fig. 5f). This result was confirmed by confocal LSM, which showed that GFP-WIPI3 co-localized with endogenous TSC2 and LAMP2 in starved cells (Fig. 5g; Supplementary Fig. 5a). In addition, WIPI3 also localized at nascent autophagosomes upon starvation (see also Fig. 1d–f), and co-localizing GFP-WIPI3 and endogenous WIPI2 puncta were negative for LAMP2 (Fig. 5h; Supplementary Fig. 5b). In fed cells GFP-WIPI3 also co-localized with LAMP2, however, this co-localization was more apparent upon lysosomal inhibition by bafilomycin A1 (Supplementary Fig. 5c, upper panel) and was sensitive to PI3K inhibition (Supplementary Fig. 5c, lower panel). TSC2 KD significantly reduced the number of GFP-WIPI3- (Fig. 5i), GFP-WIPI1- and GFP-LC3-puncta-positive cells (Supplementary Fig. 5d). As the TSC complex displaces mTOR from the lysosomal surface for TORC1 inhibition, we anticipated that WIPI3 would not localize to LAMP2-positive lysosomes/late endosomes upon TORC1 stimulation, as indicated by site-specific ULK phosphorylation (S757 in mice and S758 in humans), mTOR autophosphorylation (Fig. 5j) and lysosomal positioning (Fig. 5k; Supplementary Fig. 5e). Indeed, upon TORC1 stimulation, GFP-WIPI3 was not detected at lysosomes (LAMP2-positive lysosomes/late endosomes) positive for endogenous mTOR (Fig. 5k; Supplementary Fig. 5e). These data suggest that WIPI3 functions in association with the TSC complex to regulate mTOR activity in the lysosomal compartment.

To further examine the possibility of an additional function of WIPI3 at nascent autophagosomes, we also assessed the WIPI3-FIP200 interaction identified in the MS analysis (Fig. 3). GFP-WIPI3 clearly associated with endogenous FIP200 in MEFs (Fig. 6a). Upon starvation, GFP-WIPI3 strongly co-localized with endogenous FIP200, myc-tagged WIPI2B (Fig. 6b; Supplementary Fig. 6a) and endogenous p62 (Fig. 6c; Supplementary Fig. 6b). FIP200 KD (Fig. 6d) significantly reduced the number of GFP-WIPI3-puncta-positive cells present upon starvation (Fig. 6e) and reduced the number of GFP-WIPI1 (Fig. 6f) and GFP-LC3 (Fig. 6g) puncta.

Moreover, as FIP200 has also been reported to have additional functions and to interact with TSC1, we examined whether WIPI3 was found together with FIP200 in the lysosomal compartment. Indeed, GFP-WIPI3 co-localized with endogenous FIP200 and LAMP2-positive lysosomes/late endosomes upon starvation (Fig. 6h, upper panel; Supplementary Fig. 6c), which was abolished upon amino acid-induced TORC1 stimulation (Fig. 6h, lower panel; Supplementary Fig. 6c). These data argue that WIPI3 interacts with FIP200 at nascent autophagosomes and at LAMP2-positive lysosomes/late endosomes.

**The WIPI4 complex contains ATG2, AMPK and ULK1**. WIPI4 is considered to function, together with ATG2, downstream of LC3 at nascent autophagosomes[24]. Our MS analysis of WIPI4 protein interaction partners highlighted an association between WIPI4 and ATG2 (ATG2A, ATG2B) (Fig. 3), which we confirmed by co-immunoprecipitation (Fig. 7a). By mutating individual amino acids in WIPI4 that are conserved in human WIPI proteins[32] (Supplementary Fig. 1a) we identified two critical residues in WIPI4, N15 and D17, that confer ATG2A binding (Fig. 7b, Supplementary Fig. 7a). A phospholipid-binding mutant of WIPI4 (WIPI4-R232A/R233A, Supplementary Fig. 7a) was able to bind to ATG2A, in line with the previous suggestion that the phospholipid-binding site in all WIPI proteins is positioned on one side of the β-propeller, whereas the capacity for specific protein–protein interactions is conferred by homologous residues on the opposite site[32].

Next, we silenced WIPI4 and ATG2 and found that their combined KD (Fig. 7c) and the KD of ATG2 alone (Fig. 7d) provoked massive accumulations of WIPI1-positive membranes (Fig. 7c,d). Using automated high-throughput image analysis, we verified this result, as the size and number of both GFP-WIPI1 and GFP-LC3 (Fig. 7e) punctate structures significantly increased upon ATG2 KD. This observation suggests that the WIPI4-ATG2 complex at nascent autophagosomes is required for controlling their size downstream of the effects of LC3, which is in line with the appearance of cup-shaped double-membrane structures resembling elongated phagophore formation sites upon WIPI4 KD (Fig. 2b) and the reduced number of GFP-WIPI4-puncta-positive cells upon WIPI1 KD (Fig. 4e) and WIPI2 KD (Fig. 4f).

Our MS analysis of WIPI4-interacting proteins further revealed that AMPK likely interacts with WIPI4 (Fig. 3, Supplementary Data 2), which we assessed using GFP-WIPI4, and GFP-WIPI2B as well as GFP alone as negative controls (Fig. 7f). In fed conditions, we detected endogenous AMPK in the WIPI4-ATG2A immunoprecipitate, but we found less in starved conditions (Fig. 7f). Moreover, the WIPI4-ATG2A complex also contained endogenous ULK1 (Fig. 7g), which we did not identify in the MS analysis but which is known to interact with AMPK and to be used as an AMPK substrate upon

**Figure 4 | WIPI1 assists WIPI2 in recruiting ATG16L for LC3 lipidation.** (**a**) U2OS cells transiently expressing GFP-WIPI1, GFP-WIPI2B, GFP-WIPI2D, GFP-WIPI3, GFP-WIPI4 or GFP alone were starved for 3 h and analysed by anti-GFP immunoprecipitation followed by anti-GFP or anti-ATG16L immunoblotting. The asterisk in the right panel (GFP-IP) represents a nonspecific band. Endogenous ATG16L isoforms co-precipitated with GFP-WIPI1, GFP-WIPI2B and GFP-WIPI2D as indicated in the right panel. (**b,c**) G361 cells stably expressing shRNA targeting WIPI2 (**b**, shWIPI2), WIPI1 (**c**, shWIPI1) or the non-targeting control (shControl) were fed (F) or starved (S) for 3 h, and endogenous ATG16L was detected using anti-ATG16L/IgG-Alexa Fluor 488 antibodies and fluorescence microscopy. Mean percentages of ATG16L-puncta-positive cells (up to 369 individual cells per condition, n = 3) are presented. (**d**) Monoclonal U2OS cells stably expressing GFP-WIPI1 and shWIPI2 or shControl were starved (S) for 3 h, and the percentage of cells displaying an accumulation of perinuclear GFP-WIPI1 was calculated using fluorescence microscopy. Mean values (400 cells per condition, n = 4) are presented. (**e,f**) G361 cells stably expressing shRNAs targeting WIPI1 (**e**, shWIPI1) or WIPI2 (**f**, shWIPI2) were fed (F) or starved (S) for 3 h and immunostained using anti-WIPI4/IgG-Alexa Fluor 488 antibodies. Mean percentages of WIPI4-puncta-positive cells (up to 340 cells per condition, n = 3) are presented. (**g**) G361 cells stably expressing shRNAs targeting WIPI3 (shWIPI3), WIPI4 (shWIPI4) or the non-targeting shRNA control (shControl) were fed (F) or starved (S) for 3 h and immunostained using anti-ATG16L/IgG-Alexa Fluor 488 antibodies for analysis by fluorescence microscopy (left panel). Mean percentages of ATG16L-puncta-positive cells (up to 387 cells per condition, n = 3) are presented (right panel). (**h,i**) ATG5 wild-type (WT) or knockout (KO) mouse embryonic fibroblasts (MEFs) transiently expressing GFP-WIPI1, GFP-WIPI2B, GFP-WIPI3 or GFP-WIPI4 were analysed by confocal LSM, and images from ATG5 WT MEFs (**h**, upper panel) were processed using Volocity to generate 3D-reconstruction fly-through movies (Supplementary Movies 5–8), from which still images are presented (**h**, lower panel). Mean percentages of GFP-WIPI puncta-positive cells (300 cells, n = 3). Mean ± s.d.; heteroscedastic t-testing; P values: *P < 0.05, ns: not significant. Scale bars: 20 μm. Statistics and source data can be found in Supplementary Data 1.

starvation[68]. By overexpressing HA-tagged ULK1, we confirmed the association between GFP-WIPI4 and HA-ULK1 in fed conditions (Supplementary Fig. 7b). These results demonstrate that WIPI4-ATG2A interacts with AMPK-ULK1 in fed conditions but less upon AMPK activation. Using the pharmacological compound AICAR (5-aminoimidazole-4-carboxamide ribonucleotide), which specifically activates AMPK, the interaction of WIPI4-ATG2A and AMPK-ULK1 was further reduced (Fig. 7h, Supplementary Fig. 7c), which is underlined by quantifying the abundance of

co-immunoprecipitated AMPK with GFP-WIPI4 (Fig. 7i). As AMPK is activated upon glucose starvation, we deprived cells of glucose, either alone or in combination with glutamine (Fig. 7j). In line with the results presented above, we detected AMPK in complex with WIPI4-ATG2A in fed conditions but reduced upon glucose starvation (Fig. 7j). Using overexpressed AMPK subunits α1-γ1, the association between WIPI4 and AMPK was further confirmed in fed conditions (Fig. 7k, Supplementary Fig. 7d).

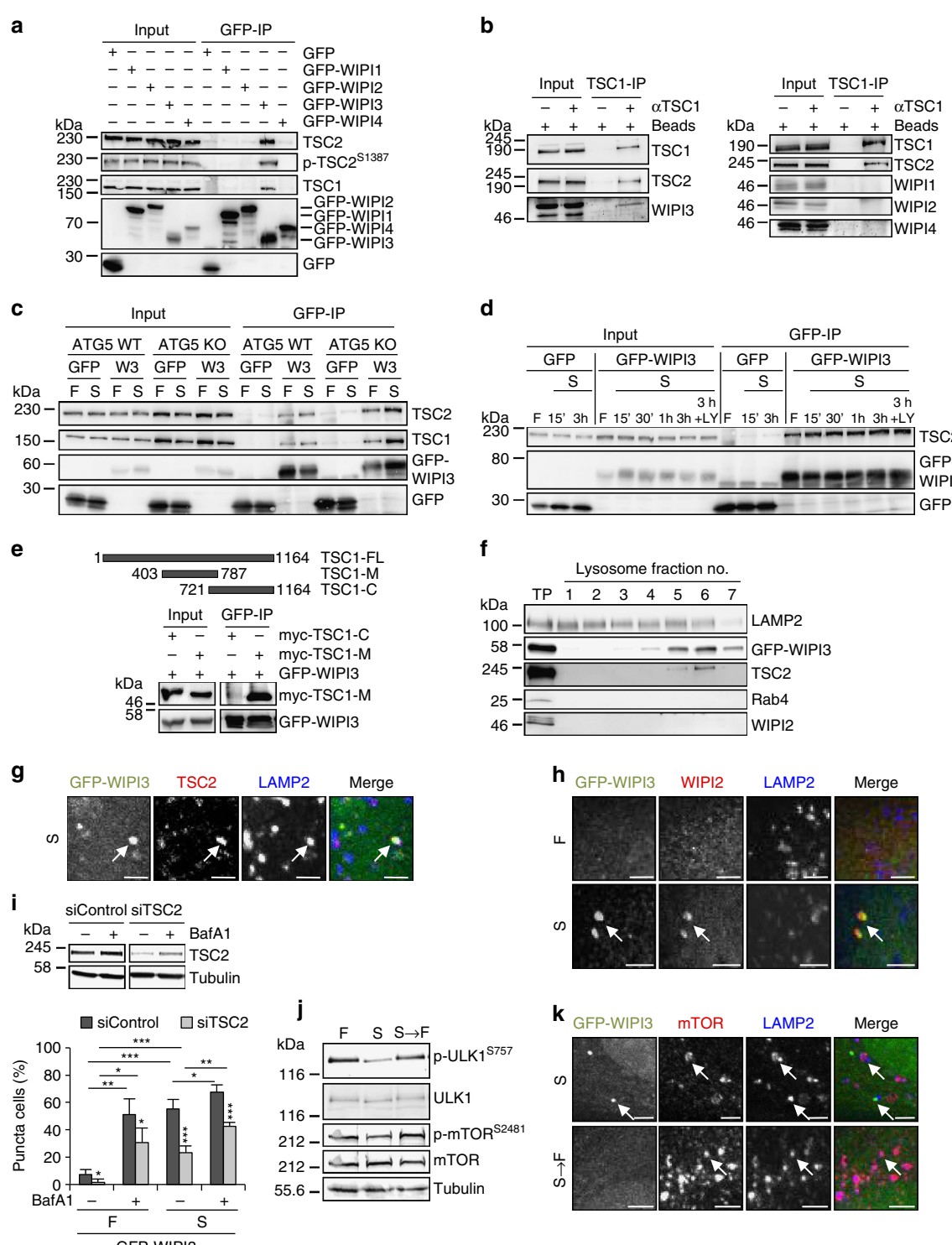

**WIPI4 is stimulated by AMPK, NUAK2 and BRSK2.** To address the functional relevance of AMPK in the control of WIPI proteins, we screened a lentiviral short hairpin RNA (shRNA) library, individually targeting all human protein kinases, and identified the regulatory AMPKγ subunit and two AMPK-related protein kinases, NUAK2 and BRSK2, that together with AMPK, are functionally and physically regulated as a network through LKB1 (ref. 69) (Fig. 8a; Supplementary Fig. 8a,b; see Methods section for details). Based on this result, we knocked down AMPKα1/2, AMPKγ1, NUAK2 and BRSK2 or their upstream kinase, LKB1, using transient short interfering RNA (siRNA) transfections (Fig. 8b) in the following experiments. Strikingly, we found a significantly reduced number of GFP-WIPI4-puncta-positive cells upon amino-acid starvation (Fig. 8c). This result suggests that LKB1/AMPK signalling upon amino-acid starvation is, in addition to its reported effects leading to TORC1 inhibition and ULK1 stimulation, also directed towards WIPI4-ATG2, which we identified to be associated with AMPK-ULK1 in fed but less in starved conditions (Fig. 7). In line with this observation, we found that glucose and glucose plus glutamine starvation, which activated AMPK as expected (Fig. 8d), provoked a significant increase in GFP-WIPI4 punctate structures (Fig. 8e), which was not the case when GFP-WIPI1 (Fig. 8f) punctate structures were assessed by automated imaging, supporting the findings of previous reports[35,36]. Further, using our panel of WIPI4 mutants (Supplementary Fig. 7a) we found that WIPI4 D113A was impaired in binding to AMPK (Supplementary Fig. 8c). Employing the GFP-WIPI4 D113A mutant (Fig. 8g,h), we found that this mutant did not respond to glucose starvation with an increase in the formation of punctate structures when compared with wild-type GFP-WIPI4 (Fig. 8g). Moreover, whereas GFP-WIPI4-puncta-positive structures significantly decreased in AMPK KD conditions (Fig. 8c,h), GFP-WIPI4 D113A punctate structures did not (Fig. 8h).

## Discussion

The human WIPI2 β-propeller protein (WIPI2B and WIPI2D) functions as an essential PtdIns3P effector at nascent autophagosomes[26]; however, the contributions of the other WIPI proteins, WIPI1 and WIPI4, in autophagy remained obscure, and WIPI3 was entirely uncharacterized[24]. Here we provide evidence that all human WIPI proteins function as scaffold building units that interlink the control of autophagy with the formation of functional autophagosomes. We used a combined proteomics and kinome screening approach and identified the WIPI interactome and the associated regulatory protein kinases. Our study shows that WIPI proteins function as scaffolds with distinct associations to both critical regulators of autophagy and each other.

We show that WIPI1 (refs 32,70) interacts with WIPI2 (WIPI2B and WIPI2D[31]) and that it supports the PtdIns3P-effector function of WIPI2 (refs 25,26) in autophagy, which is the recruitment of the ATG16L complex for LC3 lipidation[26]. Although WIPI2 silencing more prominently interferes with autophagosome formation, fewer autophagosomes form and less proteolysis occurs in the absence of WIPI1, which is consistent with a supporting role of WIPI1 for WIPI2.

At nascent autophagosomes, WIPI1-WIPI2 co-localizes with WIPI4 (refs 32,34), and in co-immunoprecipitation and proteomics analyses, we also found that WIPI1, WIPI2 and WIPI4 co-associate. In addition, WIPI1, WIPI2 and WIPI4 share NudC[60–62] as a novel partner protein. Moreover, we found that WIPI4 directly associates with ATG2 (refs 40,43,71–73) (via N15/D17) and that this complex also contains AMPK and ULK1 in fed conditions. Upon starvation and AMPK activation, WIPI4-ATG2 dissociates from AMPK and ULK1 and localizes at nascent autophagosomes, potentially supporting further autophagosome maturation. WIPI4, ATG2 or combined WIPI4/ATG2 silencing arrests autophagosome formation and increases the accumulation of early and intermediate structures, arguing for a role of WIPI4-ATG2 in controlling the extent of autophagosomal membrane formation. This function is initiated by LKB1-mediated AMPK activation[15,69], demonstrating that AMPK controls not only PtdIns3P production via ULK1 activation[68] but also appropriate autophagosome formation via WIPI4-ATG2. The AMPK link with WIPI4-ATG2 is further supported by our finding that glucose starvation triggers an increase in WIPI4 localization at nascent autophagosomes, which others and we did not observe when analysing WIPI1 and WIPI2 (refs 36,37). This finding is supported by the results of our kinome screening, which identified AMPK and the AMKP-related kinases NUAK2 and BRSK2, all of which function downstream of LKB1 (ref. 69) and stimulate the localization of WIPI4 to nascent autophagosomes.

LKB1-AMPK signalling further triggers the activation of TSC1-TSC2 (refs 10–12), which inhibits mTOR activity at lysosomes[13,14]. Interestingly, analysis of the WIPI3 interactome revealed a specific interaction of WIPI3 with the TSC complex, and we demonstrated that WIPI3 binds to the TSC complex when TSC2 is phosphorylated at serine 1,387 (S1387), an AMPK target site[16]. Moreover, we detected both WIPI3 and the TSC complex in the lysosomal compartment upon starvation but not when mTOR was activated. The region in TSC1 to

**Figure 5 | WIPI3 interacts with the TSC complex.** (**a**) Stable GFP-WIPI1, GFP-WIPI2B, GFP-WIPI3 or GFP-WIPI4 U2OS cells were starved (3 h), and subjected to anti-GFP immunoprecipitation and anti-TSC2, anti-phospho-TSC2 (S1387), anti-TSC1 or anti-GFP immunoblotting. (**b**) U2OS cells were analysed by anti-TSC1 immunoprecipitation (TSC1-IP), anti-TSC1, anti-TSC2 and anti-WIPI1 (right panel), anti-WIPI2 (right panel), anti-WIPI3 (left panel) or anti-WIPI4 (right panel) immunoblotting. (**c**) ATG5 WT or KO MEFs expressing GFP-WIPI3 (W3) or GFP were subjected to anti-GFP immunoprecipitation and anti-TSC1, anti-TSC2 and anti-GFP immunoblotting. Conditions (3 h): fed (F) or starved (S). (**d**) Stable GFP-WIPI3 or GFP U2OS cells were analysed by anti-GFP immunoprecipitation and anti-TSC2 and anti-GFP immunoblotting. Conditions (15 min. to 3 h): fed (F), starved (S), starved with LY294002 (S + LY). (**e**) Myc-tagged TSC1 fragments (TSC1-M, TSC1-C; full length: TSC-FL) were expressed in stable GFP-WIPI3 U2OS cells and subjected to anti-GFP immunoprecipitation and anti-GFP or anti-myc immunoblotting. (**f**) Lysosomal fractions (no. 1–7; total protein control: TP) from stable GFP-WIPI3 U2OS cells (BafA1, 3 h) were immunoblotted using anti-GFP, anti-LAMP2, anti-TSC2, anti-WIPI2 or anti-Rab4 antibodies. (**g**) Stable GFP-WIPI3 U2OS cells were starved (3 h) and immunostained with anti-TSC2/IgG-Alexa Fluor 546 and anti-LAMP2/IgG-Alexa Fluor 633 antibodies for confocal LSM. (**h**) Stable GFP-WIPI3 U2OS cells were immunostained with anti-Lamp2/IgG-Alexa Fluor 633 and anti-WIPI2/IgG-Alexa Fluor 546 for confocal LSM. Conditions (3 h): fed (F), starved (S). (**i**) Stable GFP-WIPI3 U2OS cells with siControl or siTSC2 were fed (F) or starved (S) with or without BafA1 (3 h). Upper panel: anti-TSC2 immunoblotting, lower panel: GFP-WIPI3 puncta assessment (up to 493 cells per condition, $n = 4$). (**j**) U2OS cells were subjected to immunoblotting using anti-phospho-mTOR (S2481), anti-mTOR, anti-phospho-ULK1 (S757), anti-ULK1 and anti-tubulin antibodies. Treatments: fed (F, 4 h), starved (S, 4 h), starved (3 h) and fed (1 h) (S→F). (**k**) Stable GFP-WIPI3 U2OS cells were immunostained with anti-mTOR/IgG-Alexa Fluor 546 and anti-LAMP2/IgG-Alexa Fluor 633 antibodies. Treatments: starved (S, 2 h), starved (2 h) and fed (1 h) (S→F). Co-localizations are indicated with arrows (**g,h,k**). Supplementary Material is available: Supplementary Fig. 5. Statistics and source data: Supplementary Data 1. Mean ± s.d.; heteroscedastic $t$-testing; $P$ values: $*P < 0.05$, $**P < 0.01$, $***P < 0.001$. Scale bars: 3 μm.

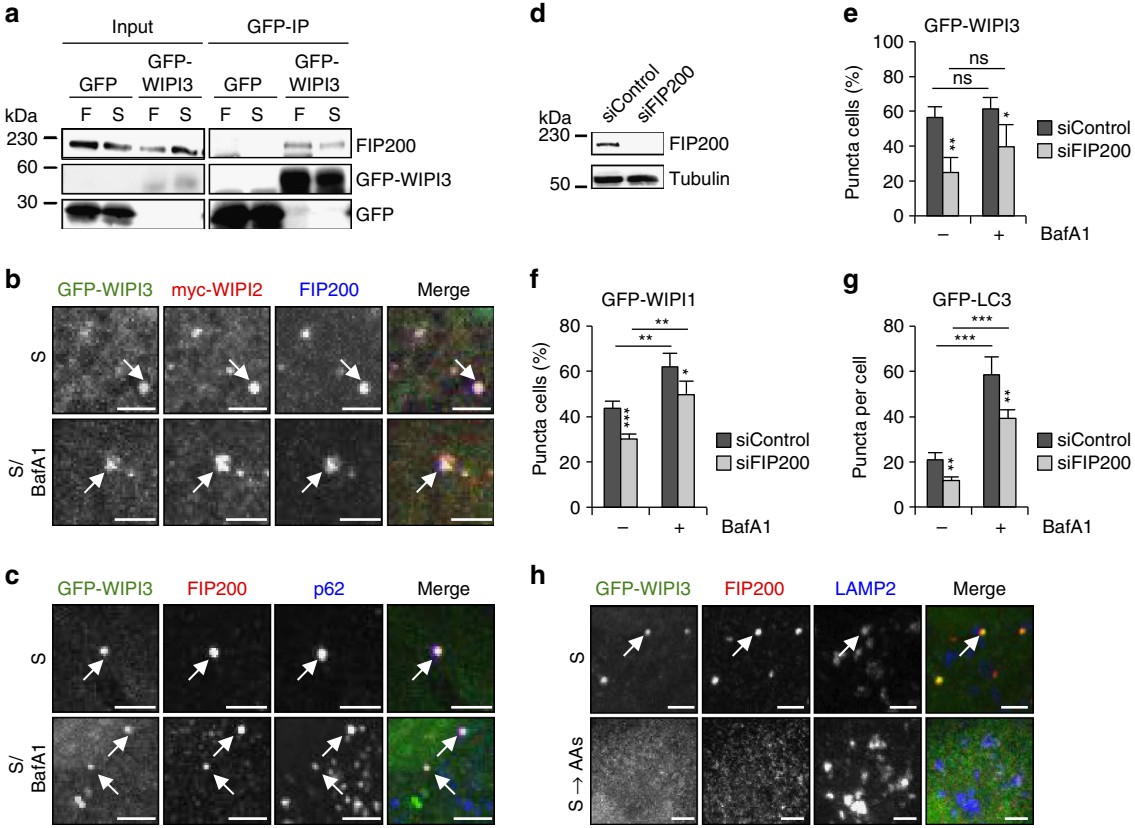

**Figure 6 | WIPI3 interacts with FIP200 at nascent autophagosomes and at LAMP2-positive lysosomes/late endosomes.** (**a**) Mouse embryonic fibroblasts transiently expressing GFP-WIPI3 or GFP alone were fed (F) or starved (S) for 3 h and analysed by immunoprecipitation using anti-GFP antibodies and anti-FIP200 or anti-GFP immunoblotting. (**b**) U2OS cells stably expressing GFP-WIPI3 were transfected with myc-WIPI2B and starved (S) in the absence or presence of BafA1 for 3 h. Cells were immunostained with anti-myc/IgG-Alexa Fluor 546 or anti-FIP200/IgG-Alexa Fluor 633 antibodies and visualized by confocal LSM. (**c**) U2OS cells stably expressing GFP-WIPI3 were starved (S) in the absence or presence of BafA1 for 3 h, immunostained with anti-FIP200/IgG-Alexa Fluor 546 and anti-p62/IgG-Alexa Fluor 633 antibodies and visualized by confocal LSM. (**d,e**) U2OS cells stably overexpressing GFP-WIPI3 were transfected with control siRNA or FIP200 siRNA. Subsequently, the cells were starved for 3 h in the absence or presence BafA1. FIP200 downregulation was confirmed by immunoblotting using anti-FIP200 and anti-tubulin antibodies (**d**). Mean percentages of GFP-WIPI3-puncta-positive cells (up to 481 cells per condition, $n = 4$) are presented (**e**). (**f**) U2OS cells stably overexpressing GFP-WIPI1 were transfected with control siRNA or FIP200 siRNA. Subsequently, the cells were starved for 3 h with nutrient-free medium in the absence or presence BafA1. Using automated image acquisition and analysis, the mean percentages of GFP-WIPI1-puncta-positive cells (up to 5,856 cells per condition, $n = 4$) were calculated. (**g**) In parallel, this experiment was conducted using GFP-LC3-expressing U2OS cells and mean numbers of GFP-LC3 puncta per cell (up to 4,798 cells per condition, $n = 4$) calculated. (**h**) U2OS cells stably expressing GFP-WIPI3 were starved (S) for 2 h or starved and replenished with amino acids for 1 h (S→AAs). Subsequently, cells were immunostained with anti-FIP200/IgG-Alexa Fluor 546 and anti-LAMP2/IgG-Alexa Fluor 633 antibodies and visualized by confocal LSM. Image magnifications are presented (**b,c,h**) and co-localizations indicated with white arrows. Supplementary Material is available: Supplementary Fig. 6. Statistics and source data can be found in Supplementary Data 1. Mean ± s.d.; heteroscedastic $t$-testing; $P$ values: $*P < 0.05$, $**P < 0.01$, $***P < 0.001$, ns: not significant. Scale bars: 3 μm.

which WIPI3 binds overlaps with the reported binding site for FIP200 (ref. 66). In line with this finding, we also identified and verified a specific interaction between WIPI3 and FIP200, from which we suggest that WIPI3 functions as a scaffold between the TSC complex and FIP200. In support of this idea, we found that WIPI3 co-localizes with FIP200 in the lysosomal compartment, a reported localization of FIP200 with unknown consequences[74]. The data presented here suggest that WIPI3 in complex with TSC1-TSC2 and FIP200 co-regulate mTOR activity; in line with this hypothesis, WIPI3 was found to be absent from lysosomes when mTOR was present. Our results further emphasize that FIP200 performs numerous roles in autophagy, acting as part of the ULK1 complex[63], together with WIPI3 and the TSC complex at lysosomes, and at nascent autophagosomes, where direct binding between FIP200 and ATG16L has been reported[75,76]. Upon starvation, WIPI3 also translocates to the nascent autophagosome, where it co-localizes with FIP200 and all other WIPI proteins and ATG members examined. WIPI3 deficiency arrests autophagosome formation, indicating that WIPI3 also has a role in the formation of functional autophagosomes.

Taken together, our data reveal the WIPI interactome and suggest that the LKB1-AMPK regulatory network directly or indirectly regulates the scaffold functions of the four human WIPI proteins. Our work provides a framework for further studies concentrating in more detail on the various new aspects presented here.

## Methods

**Antibodies.** The following primary antibodies were used in this study: AMPKα (Cell Signaling; #2,532; WB 1:1,000), AMPKγ1 (Cell Signaling; #4,187; WB 1:1,000), Phospho-AMPKα (Thr172) (Cell Signaling; #2,531; WB 1:1,000), ATG12 (Abgent; AP1816a; IF 1:25), ATG16L (CosmoBio; CAC-TMD-PH-ATG16L; WB

1:1,000; IF 1:25), CaMKKα (Santa Cruz; sc-136280; WB 1:1,000), FIP200 (Cell Signaling; #12,436; WB 1:1,000, IF 1:100), GAPDH (Acris; ACR001P; WB 1:50,000), GFP (Roche; #11814460001; WB 1:1,000), HA.11 (Biolegend; MMS-101R; WB 1:1,000), LAMP2 (Santa Cruz; sc-18,822; WB 1:1,000; IF 1:50), LC3 (Nanotools; 0231-100/LC3-5F10; WB 1:1,000, IF 1:5), LKB1 (Cell Signaling; #3,047; WB 1:1,000), c-myc (SantaCruz; sc-40, sc-789; WB 1:1,000; IF 1:200), NudC (Abgent; AT3125a; WB 1:1,000), p62 (Santa Cruz; sc-28,359; WB 1:1,000; IF 1:50), Rab4 (BD Transduction Laboratories; 610,888; WB 1:1,000), mTOR (Cell Signaling; #2,983; WB 1:1,000; IF 1:400), Phospho-mTOR (Ser2481) (Cell Signaling; #2,974; WB 1:1,000), Hamartin/TSC1 (Cell Signaling; #6,935, #4,906; WB 1:1,000), Tuberin/TSC2 (Cell Signaling; #4,308; WB 1:1,000; IF 1:800), Phospho-Tuberin/TSC2 (Ser1387) (Cell Signaling; #5,584; WB 1:1,000), αTubulin (Sigma-Aldrich; T5168; WB 1:50,000), γTubulin (Sigma-Aldrich; T3559; WB 1:50,000), ULK1 (Cell Signaling; #8,054; WB 1:1,000), Phospho-ULK1 (Ser757) (Cell Signaling; #6,888; WB 1:1,000), WIPI1 (Abnova; H00055062-M02; WB 1:1,000; IF 1:25), WIPI2 (Abgent; AP9559b; WB 1:250; IF 1:25), WIPI3/WDR45L

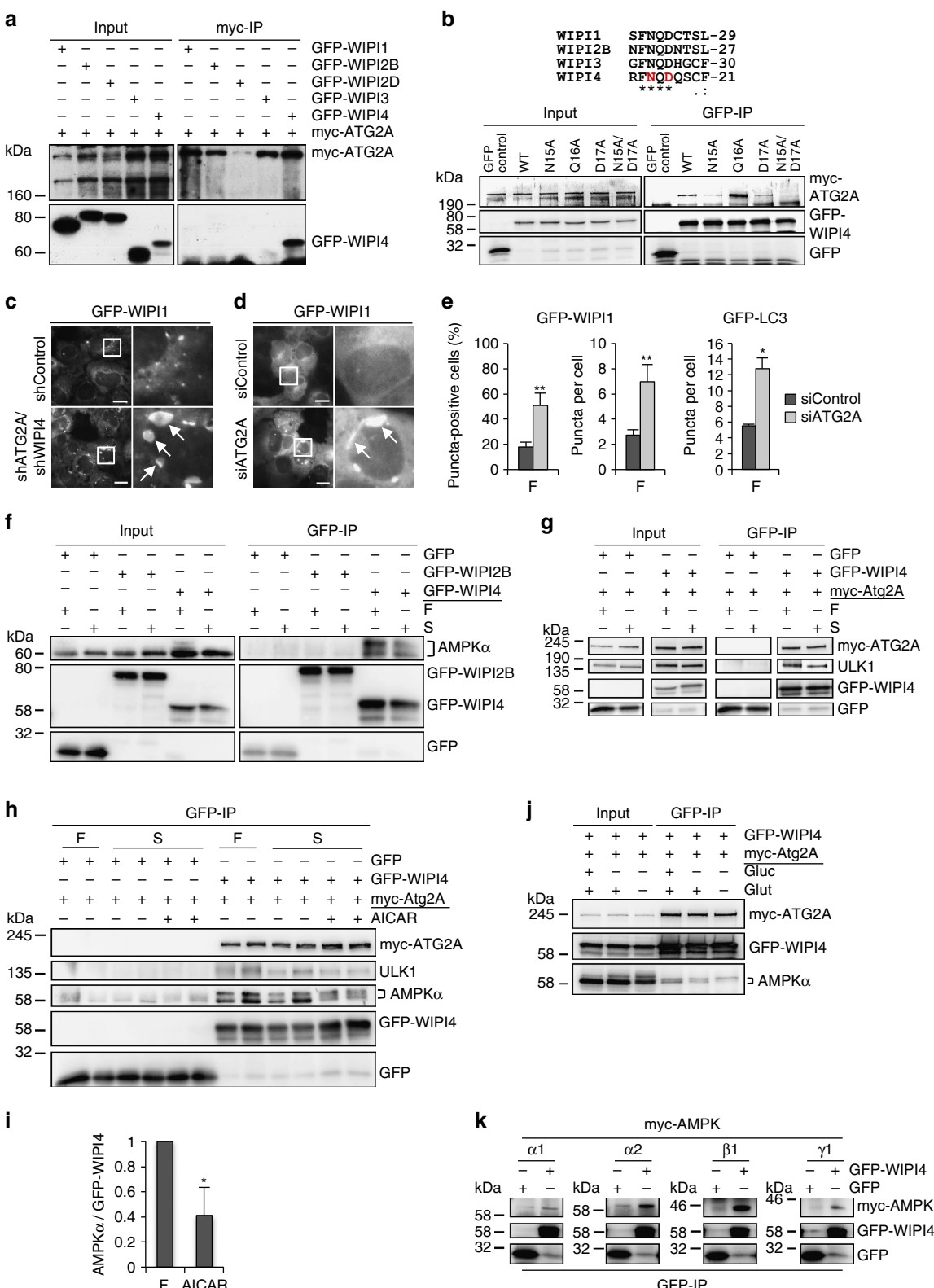

(Sigma-Aldrich; HPA017886; IF 1:25), WIPI3 (Santa Cruz; sc-514194; WM 1:500), WIPI4/WDR45 (Abnova; H00011152-B01P; WB 1:200; IF 1:25).

For immunofluoresence the following secondary antibodies were purchased from Molecular Probes/Life Technologies: Alexa Fluor 488 goat anti-rabbit IgG (A-11008; IF 1:200), Alexa Fluor 488 goat anti-mouse IgG (A-11001; IF 1: 200), Alexa Fluor 546 goat anti-rabbit IgG (A-11010; IF 1:200), Alexa Fluor 546 goat anti-mouse IgG (A-11003; IF 1:200), Alexa Fluor 633 goat anti-rabbit IgG (A-21070; IF 1:200), Alexa Fluor 633 goat anti-mouse IgG (A-21050; IF 1:200).

For immunoblotting the following secondary antibodies were used in this study: anti-rabbit IgG, horseradish peroxidase (HRP)-linked (GE Healthcare; NA934V; WB 1:10,000), anti-mouse IgG, HRP-linked (GE Healthcare; NA931V; WB 1:10,000), anti-rabbit IgG, HRP-linked (Cell Signaling; #7,074; WB 1:10,000), anti-mouse IgG, HRP-linked (Cell Signaling; #7,076; WB 1:10,000).

**Short interfering RNAs.** The following siRNAs were used in this study: WIPI2 (EHU095551), WIPI3/WDR45L (EHU109971), WIPI4/WDR45 (EHU023161) and the control esiRNA targeting FLUC (EHUFLUC) were purchased from Sigma-Aldrich. TSC2/Tuberin (#6,476) and control siRNA (#6,568) were purchased from Cell Signaling. ATG2A (sc-96,345), AMPKα1/2 (sc-45,312), AMPKγ1 (sc-38,929), BRSK2 (sc-96,315), CaMKKα (sc-29,904), LKB1 (sc-35,816), NUAK2/SNARK (sc-88,608), NudC (sc-88,034), TSC1/hamartin (sc-37,437), FIP200/RB1CC1 (sc-38,211), and control siRNA-A (sc-37,007) were purchased from Santa Cruz.

**Complementary DNA (cDNA) constructs.** GFP-WIPI1, GFP-WIPI2B, GFP-WIPI2D, GFP-WIPI3S, GFP-WIPI4, myc-WIPI1, myc-WIPI2B, as well as myc-WIPI2D were previously reported[31,32]. Myc-ATG2A, myc-ATG14 and myc-DFCP1 were also previously reported[31,43]. Cloning of GFP-WIPI3 was conducted by using the following oligonucleotides: forward primer: 5′-agagagctc gagctatggattccggcgccatgaac-3′, reverse primer: : 5′-agagagggatcctcacagcttgtcatcggt cag-3′. This WIPI3 cDNA sequence encoded an N-terminal extension (Supplementary Fig. 1a, Supplementary Note, GeneBank accession number KX434429) when compared with the original sequence (AY691427.1) (ref. 32).

The GFP-LC3 (refs 29,48) plasmid was kindly provided by Tamotso Yoshimori, Osaka University, Japan. Myc-AMPKα1 was generated by subcloning the human AMPKα1 cDNA (Addgene plasmid #20,595: pWZL Neo Myr Flag PRKAA1 was a gift from William Hahn and Jean Zhao[77]) into pCMV-Tag3b (XhoI/ApaI) using the following oligonucleotides: forward primer: 5′-agagagctcgagatggcgacagccgagaag cag-3′, reverse primer: 5′-agagaggggccccaattgtgcaagaatttaattag-3′. Myc-AMPKα2 was generated by subcloning human AMPKα2 cDNA (Addgene plasmid #31,654: pECE-HA-AMPKalpha2 wt was a gift from Anne Brunet[78]) into pCMV-Tag3b (BamHI/XhoI) using the following oligonucleotides: forward primer: 5′-agagagg gatccatggctgagaagcagaagcac-3′, reverse primer: 5′-agagagctcgagcaaacgggctaaagtag taatc-3′. Myc-AMPKβ1 was generated by subcloning human AMPKβ1 cDNA (Addgene plasmid #31,666: pECE-HA-AMPKβ1 wt was a gift from Anne Brunet[78]) into pCMV-Tag3b (XhoI/ApaI) using the following oligonucleotides: forward primer: 5′-agagagctcgagggcaataccagcagtgagcg-3′, reverse primer: 5′-agagag gggcccgtttaaactatgggcttgtataac-3′. Myc-AMPKγ1 was generated by subcloning human AMPKγ1 cDNA (Addgene plasmid #30,308: pFLAG-Cherry-N1-PRKAG1 was a gift from Jay Brenman[79]) into pCMV-Tag3b (BamHI/XhoI) using the following oligonucleotides: forward primer: 5′-agagagggatccatggagacggtcatttctt cag-3′, reverse primer: 5′-agagagctcgagctagggcttcttctcctccacc-3′. Human HA-hULK1 was a gift from Do-Hyung Kim[64] and received from Addgene (plasmid #31,963). Myc-TSC1 was generated by subcloning TSC1 cDNA (Addgene plasmid #19,955: pcDNA3-myc3-Tsc1 was a gift from Yue Xiong[80]) into pCMV-Tag3b (SrfI/ApaI)

using the following oligonucleotides: forward primer: 5′-agagaggcccgggcatggcc caacaagcaaatg-3′, reverse primer: 5′-gagagagggcccttattagctgtgttcatgatgatgag-3′. Myc-TSC1-C was generated by subcloning TSC1 cDNA into pCMV-Tag3b (SrfI/ApaI) using the following oligonucleotides: forward primer: 5′-agagaggcccggg ccgcaaggtgatcaaagc-3′, reverse primer 5′-gagagggccccttattagctgtgttcatgatgatgag-3′. Myc-TSC1-M was generated by subcloning TSC1 cDNA into pCMV-Tag3b (SrfI/ApaI) using the following oligonucleotides: forward primer: 5′-agagaggcccg ggccattcggatgactacgtgc-3′, reverse primer: 5′-agagaggggcccttactctcggtcatgctgcag-3′.

Construct integrities were confirmed by restriction enzyme digests and automated DNA sequencing (LGC Genomics GmbH).

**Site-directed mutagenesis.** The generated GFP-WIPI3 encoding construct (see above) was used as a DNA template and synthesized mutant oligonucleotides (Sigma-Aldrich) for site-directed mutagenesis (Quick change II; Stratagene) to generate the following construct encoding mutant WIPI3 protein: GFP-WIPI3-R230A/R231A (forward primer: 5′-ttaatccaggaactggcagcaggatctcaagca gcc-3′, reverse primer: 5′-ggctgcttgagatcctgctgccagttcctggattaa-3′).

The GFP-WIPI4 encoding construct[32] was used as a DNA template and synthesized mutant oligonucleotides (Sigma-Aldrich) for site-directed mutagenesis (Quick change II; Stratagene) to generate the following constructs encoding mutant WIPI4 proteins.

GFP-WIPI4-N15A (forward primer: 5′-accagcctgcgtttcgcccaagaccaaagctgc-3′, reverse primer: 5′-gcagctttggtcttgggcgaaacgcaggctggt-3′), GFP-WIPI4-Q16A (forward primer: 5′-agcctgcgtttcaacgcagaccaaagctgc-3′, reverse primer: 5′-gcagctt tggtctgcgttgaaacgcaggct-3′), GFP-WIPI4-D17A (forward primer: 5′-tgcgtttcaaccaa gcccaaagctgcttttgc-3′, reverse primer: 5′-gcaaaagcagctttgggcttggttgaaacgca-3′), GFP-WIPI4-E55A (forward primer: 5′-gcatgggcttggtggcgatgctgcaccgctc-3′, reverse primer: 5′-gagcggtgcagcatcgccaccaagcccatgc-3′), GFP-WIPI4-R109A (forward primer: 5′-ccagtgctttctgtggccatgcgccatgac-3′, reverse primer: 5′-gtcatggcgcatggcc acagaaagcactgg-3′), GFP-WIPI4-R111A (forward primer: 5′-ctttctgtgcgccatggcccatg acaagatcg-3′, reverse primer: 5′-cgatcttgtcatgggccatgcgcacagaaag-3′), GFP-WIPI4-H112A (forward primer: 5′-tctgtgcgccatgcgcgctgacaagatcgtg-3′, reverse primer: 5′-ca cgatcttgtcagcgcgcatgcgcacaga-3′), GFP-WIPI4-D113A (forward primer: 5′-tgcgcat gcgccatgccaagatcgtgatcg-3′, reverse primer: 5′-cgatcacgatcttggcatggcgcatgcgca-3′), GFP-WIPI4-K114A (forward primer: 5′-cgcatgcgccatgccgacgatcgtgatcgtgc-3′, reverse primer: 5′-gcacgatcacgatcgcgtcatggcgcatgcg-3′), GFP-WIPI4-R232A/R233A (forward primer: 5′-aaactggtggagctggccgcaggcactgaccc-3′, reverse primer: 5′-gggtc agtgcctgcggccagctccaccagttt-3′), GFP-WIPI4-N15A/D17A (forward primer: 5′-acc agcctgcgtttcgcccaagcccaaagctgc-3′, reverse primer: 5′-gcagctttgggcttgggcgaaacgcag gctggt-3′).

Construct integrities were confirmed by restriction enzyme digests and automated DNA sequencing (LGC Genomics GmbH).

**Cell culture.** Human U2OS osteosarcoma cell line (ATCC; HTB-96), human G361 malignant melanoma cell line (ATCC; CRL-1424) and m5–7 MEFs (kindly provided by Noboru Mizushima, Tokyo Medical and Dental University, Japan) were cultured at 37 °C and 5% $CO_2$ in DMEM GlutaMAX (Life Technologies; 31,966) supplemented with 10% fetal calf serum (PAA; A15-043), 100 U ml$^{-1}$ penicillin/100 μg ml$^{-1}$ streptomycin (Life Technologies; 15140-122).

Doxycycline (1 μg ml$^{-1}$; Sigma-Aldrich, D9891) was added to m5–7 cells for the repression of ATG5 cDNA expression in order to obtain ATG5 knockout (KO) MEF; doxycycline was omitted from the culture medium to obtain ATG5 wild-type (WT) MEF[81].

G418 Sulfate (Life Technologies; 11811-098) was additionally added (0.6 mg ml$^{-1}$) to monoclonal U2OS cell lines stably expressing GFP-WIPI1,

**Figure 7 | WIPI4 interacts with ATG2, AMPK and ULK1. (a)** U2OS cells transiently expressing GFP-tagged WIPI proteins and myc-ATG2A were starved and analysed by anti-myc immunoprecipitation and anti-myc and anti-GFP immunoblotting. **(b)** U2OS cells expressing myc-ATG2A and GFP, GFP-WIPI4 WT or GFP-WIPI4 mutants (N15A, Q16A, D17A, N15A/D17A) were analysed by anti-GFP immunoprecipitation, and anti-myc and anti-GFP immunoblotting. Amino acids in WIPI4 conferring ATG2 binding are highlighted in red (upper panel). **(c)** Stable GFP-WIPI1 U2OS cells with shRNAs targeting WIPI4 and ATG2 (shATG2/shWIPI4) or with shControl were analysed by automated imaging. Aberrant GFP-WIPI1 accumulations are indicated. **(d,e)** Stable GFP-WIPI1 or GFP-LC3 U2OS cells were transfected with siATG2 or siControl (fed conditions, F) and images of GFP-WIPI1 cells are shown **(d)**. **(e)** GFP-WIPI1 (left, middle panel) and GFP-LC3 puncta (right panel) formation was assessed using up to 13,614 GFP-WIPI1 U2OS cells ($n = 3$) or up to 9,457 GFP-LC3 U2OS cells ($n = 3$) per condition. **(f)** Stable GFP-WIPI2B, GFP-WIPI4 or GFP U2OS cells expressing myc-ATG2A were fed (F) or starved (S) and analysed by anti-GFP immunoprecipitation and anti-GFP or anti-AMPKα immunoblotting ($n = 2$). **(g)** Stable GFP-WIPI4 or GFP U2OS cells expressing myc-ATG2A were fed (F) or starved (S) for 3 h and analysed by anti-GFP immunoprecipitation and immunoblotting (anti-GFP, anti-myc or anti-Ulk1 antibodies) ($n = 3$ with duplicates). **(h)** Stable GFP-WIPI4 or GFP U2OS cells expressing myc-ATG2A were analysed by anti-GFP immunoprecipitation and anti-GFP, anti-myc, anti-Ulk1 or anti-AMPKα immunoblotting. Conditions (3 h): fed (F), starved (S), starved with AICAR ($n = 3$ with duplicates). **(i)** The abundance of AMPK co-purifying with GFP-WIPI4 upon AICAR treatment was quantified ($n = 3$). **(j)** Stable GFP-WIPI4 cells expressing myc-ATG2A were treated (3 h) with complete medium (without FCS) lacking glucose (Gluc) or glucose/glutamine (Gluc/Glut), and analysed by anti-GFP immunoprecipitation and anti-myc, anti-AMPKα and anti-GFP immunoblotting. **(k)** U2OS cells expressing GFP or GFP-WIPI4 with myc-tagged AMPKα1, α2, β1 or γ1 were analysed by anti-GFP immunoprecipitation and anti-myc and anti-GFP immunoblotting. Additional Supplementary Material is available: Supplementary Fig. 7. Statistics and source data: Supplementary Data 1. Mean ± s.d.; heteroscedastic $t$-testing; $P$ values: *$P < 0.05$, **$P < 0.01$, ***$P < 0.001$, ns: not significant. Scale bars: 20 μm.

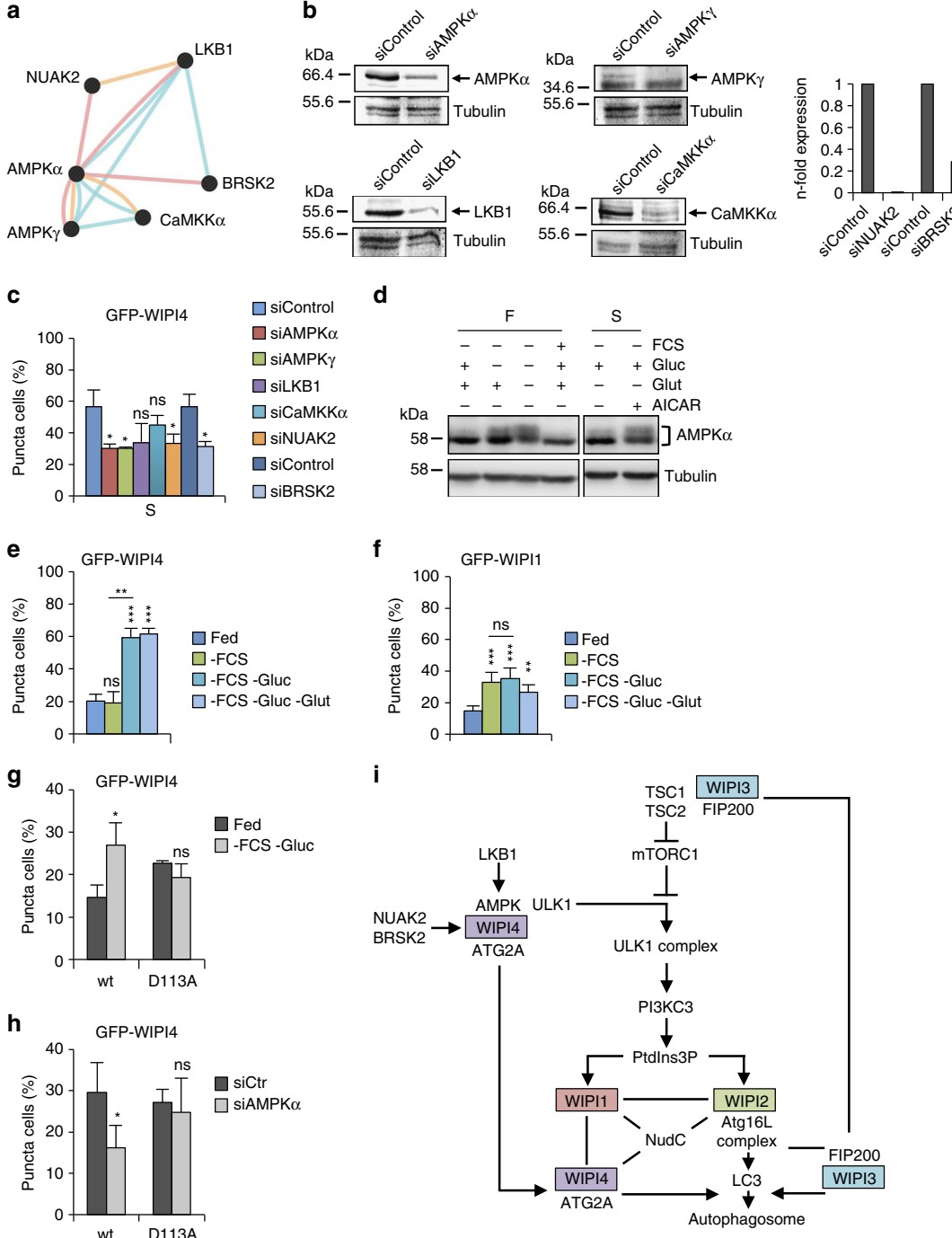

**Figure 8 | Glucose-starvation-mediated AMPK activation regulates WIPI4.** (**a**) A lentiviral-based shRNA screening approach targeting the human kinome used for the assessment of autophagy identified AMPK and the AMPK-related protein kinases NUAK2 and BRSK2, along with CaMKKα, as autophagy regulators (for details see Methods section and Supplementary Fig. 8). Reported (red) and predicted (yellow) proteins interactions, and pathway interactions (blue) are indicated (GeneMANIA). (**b**) Stable GFP-WIPI1 U2OS cells were transfected with siRNAs for AMPKα, AMPKγ, LKB1, CaMKKα, NUAK2 or BRSK2 and knock-down confirmed by immunoblotting (left panels) or quantitative RT–PCR (right panels). (**c**) Stable GFP-WIPI4 U2OS cells with siRNAs targeting AMPKα, AMPKγ, LKB1, CaMKKα, NUAK2 or BRSK2 were starved (S) for 3 h. Mean percentages of GFP-WIPI4-puncta-positive cells (300 cells per condition, $n=3$) are presented. (**d**) U2OS cells were fed (F) or starved (S) with or without glucose, glutamine or AICAR and immunoblotted using anti-AMPKα and anti-tubulin antibodies. Stable GFP-WIPI4 U2OS cells (**e**) or GFP-WIPI1 (**f**) were fed or treated with complete medium without FCS ($-$FCS), without glucose ($-$FCS $-$Gluc) or without glucose and glutamine ($-$FCS $-$Gluc $-$Glut). Mean percentages of GFP-WIPI4-puncta-positive cells (300 cells per condition, $n=3$) (**e**) and GFP-WIPI1-puncta-positive cells (up to 3,904 cells per condition, $n=6$) (**f**) are presented. (**g**) U2OS cells expressing GFP-WIPI4 WT or GFP-WIPI4 D113A mutant were fed or glucose-starved ($-$FCS $-$Gluc) for 3 h. Mean percentages of GFP-WIPI4-puncta-positive cells (300 cells per condition, $n=3$) are presented. (**h**) U2OS cells expressing GFP-WIPI4 WT or GFP-WIPI4 D113A mutant with siControl (siCtr) or siAMPKα were glucose-starved ($-$FCS $-$Gluc) for 3 h. Mean percentages of GFP-WIPI4-puncta-positive cells (500 cells per condition, $n=5$) are presented. (**i**) A predicted model for the differential contributions of human WIPI β-propeller proteins in autophagy. Statistics and source data can be found in Supplementary Data 1. Mean ± s.d.; heteroscedastic $t$-testing; $P$ values: *$P<0.05$, **$P<0.01$, ***$P<0.001$, ns: not significant.

GFP-WIPI2B, GFP-WIPI2D, GFP-LC3 or GFP alone[31,36,51] and are here referred to as stable GFP-WIPI1, GFP-WIPI2B, GFP-WIPI2D, GFP-LC3 or GFP U2OS cells.

**RNA interference.** The following lentiviral experiments have been approved (Az.: Vl-2-7534.45/11, Regierungspräsidium Tübingen, Germany) and were conducted in Biosafety Level 2 laboratory facilities. For generating a stable WIPI2 KD in U2OS and G361 cells and lentiviral packaging, lentiviral WIPI2-targeting shRNA constructs were obtained (Sigma-Aldrich; TRCN0000154324, TRCN0000153071); cell infection was conducted at a multiplicity of infection of 20 in the presence of 9 µg ml$^{-1}$ hexadimethrine bromide (Sigma- Aldrich; H9268) and selection was conducted by using 1 µg ml$^{-1}$ puromycin (Sigma-Aldrich, P9620) (ref. 82). For the stable KD of either WIPI1, WIPI3, WIPI4 or ATG2A/B the following lentiviral particles (pLKO.1) were purchased from Sigma-Aldrich and used to infect U2OS and G361 cells (in the presence of 9 µg ml$^{-1}$ hexadimethrine bromide; Sigma-Aldrich; H9268) and puromycin (1 µg ml$^{-1}$; Sigma-Aldrich; P9620) selection: WIPI1 (TRCN0000152), WIPI3 (TRCN0000131124, TRCN0000147320, TRCN0000128386), WIPI4 (TRCN0000138588, TRCN0000136607, TRCN0000134205), ATG2A (TRCN0000172342). In addition, control U2OS and G361 cell lines stably expressing scrambled shRNAs (Sigma-Aldrich; SHC002) were generated. KD events were verified by immunoblotting and quantitative PCR.

Transient KD experiments with siRNAs or esiRNAs were conducted by using Lipofectamine RNAiMAX (Invitrogen; 13778-075) according to the manufacturer's reverse transfection protocol. Briefly, siRNA (25 nM or 50 nM ) or esiRNA (200 ng) and 0.2 µl Lipofectamine RNAiMAX were diluted in 20 µl OPTI.MEM (Life Technologies; 51985-026) and the transfection solution incubated for 20 min at room temperature (RT) and added to the well of a 96-well plate. Finally, 10,000 cells (U2OS or G361) in 100 µl DMEM GlutaMAX/10%FCS were added to the transfection solution for 48 h. KD events were verified by immunoblotting and quantitative PCR.

**DNA transfection.** Transient transfections were conducted by using Promofectine (Promokine; PK-CT-2000-100) or Lipofectamine 2,000 (Invitrogen; 11668-019) according to the manufacturer's instructions. Alternatively, DEAE-Dextran (Sigma- Aldrich; D9885) was used for large-scale transient transfections and conducted in 10 cm dishes. Twelve microgram DNA was diluted in 900 µl PBS, and 90 µl DEAE-Dextran (10 mg ml$^{-1}$) was diluted in 900 µl PBS and both solutions incubated for 5 min at RT. Subsequently, the DEAE-Dextran solution was mixed with the DNA solution and DNA/DEAE-Dextran mixture incubated for 10 min at RT. The cells were washed twice with PBS and the DNA/DEAE-Dextran mixture was added to the cells. The cells were incubated for 30 min at 37 °C, 5% CO$_2$ and every 5 min the plates were rocked by hand. Subsequently, 6 ml DMEM GlutaMAX supplemented with 60 µl chloroquine (8 mM) was added to the cells and incubated at 37 °C, 5% CO$_2$. After 2.5 h, the medium was replaced with 6 ml DMEM GlutaMAX containing 10% DMSO, cells incubated for 90 s, washed with DMEM GlutaMAX and incubated for 48 h in DMEM GlutaMAX/10% FCS at 37 °C, 5% CO$_2$.

Monoclonal U2OS cell lines stably expressing GFP-WIPI3S, GFPWIPI3 or GFP-WIPI4 have been generated upon transient transfections using Lipofectamine 2,000 (Invitrogen; 11668-019) and G418 selection (0.6 mg ml$^{-1}$) (refs 31,36).

**Autophagy assay.** Cells were incubated in full (fed, F) (DMEM GlutaMAX/10%FCS) or nutrient-free starvation medium (starved, S) (EBSS; Life Technologies; 24010-043) in the presence or absence of 200 nM bafilomycin A1 (AppliChem, A7823), 233 nM wortmannin (Sigma-Aldrich, W1628), 300 nM rapamycin (Sigma-Aldrich; R0395), 100 µM LY294002 (Sigma-Aldrich; L9908) or 1 mM AICAR (Selleckchem; S1802). Unless stated otherwise, incubation periods were generally conducted for 3 h. For glucose and glucose/glutamine deprivation the following media were used: DMEM (Life Technologies; 11965-092), DMEM, no glucose (Life Technologies; 11966-025), DMEM, no glucose, no glutamine, no phenol red (Life Technologies; A14430-01).

**RNA extraction and quantitative PCR.** Total RNA was extracted by using the innuPREP RNA Mini Kit (Analytik Jena AG; 845-KS-2040250) according to the manufacturer's instruction. Standard reverse transcription (RT)–PCR was performed[83] with the following TaqMan probes (Applied Biosystems): Hs00215872_m1 (WIPI1), Hs00255379_m1 (WIPI2), Hs00750495_s1 (WIPI3/WDR45B), Hs01079049_g1 (WIPI4/WDR45), Hs00388292_m1 (NUAK2), Hs00908871_m1 (BRSK2).

**Immunoblotting.** Cells were washed with PBS, lysed in hot Laemmli buffer and boiled for 5 min at 95 °C. The chromatin was sheared with a 23 G needle. Proteins were separated by SDS–polyacrylamide gel electrophoresis, transferred to a polyvinylidene difluoride membrane (Millipore; IPVH00010) and subjected to standard antibody incubation procedures at 4 °C and ECL analysis using the enhanced chemiluminescent substrate from Millipore (WBKLS0500) or Super-Signal West Femto Maximum Sensitivity Substrate from Thermo Scientific (#34,095). ECL was detected by exposure to KODAK BioMAX Films or with the

Fusion SL instrument (Vilber Lourmat). For quantification, the Fusion Capt advance software from Vilber Lourmat was used. For all western blots uncropped western blot scans along with the locations of molecular weight/size markers and cropped sections are provided in the Supplementary Information.

**Immunopurifications.** Cells were lysed in fresh ACA lysis buffer (750 mM Aminocaproic acid, 50 mM Bis-Tris, 0.5 mM EDTA, pH 7.0, 0.1% Tween20, supplemented with Complete EDTA-free protease inhibitor cocktail (Roche; 04693132001), PhosStop phosphatase inhibitor cocktail (Roche; 04906837001). Cell lysates were mixed (vortex, 10 s), incubated on ice (20 min) and the soluble fractions obtained (centrifugation for 15 min at 20,000 g). Immunoprecipitations were conducted by incubating the soluble fractions with appropriate primary antibodies for 1 h on ice and subsequent tumbling with 70 µl protein A/G PLUS-agarose beads (SantaCruz; sc-2003) over night at 4 °C. The beads were washed three times with fresh ACA lysis buffer (see above) and the immunoprecipitates subsequently analysed by immunoblotting.

**Immunofluorescence.** Cells were grown on sterile coverslips and fixed for 10–20 min with paraformaldehyde (3.7% in PBS) for direct fluorescence (GFP) assessments. For indirect immunofluorescence fixed cells were permeabilized (PBS/0.1% Triton-X) and incubated with Image-iT FX Signal Enhancer (Life Technologies; I36933). Immunostaining with primary and secondary antibodies were conducted at 4 °C for 60 min (ref. 32). and finally, the cells were mounted on glass slides using ProLong Gold Antifade Mountant (Life Technologies; P36930).

**Low through-put fluorescence microscopy.** For general fluorescence microscopy the Axiovert 200 M microscope (Carl Zeiss GmbH) was used. For sub-cellular localization studies images were acquired by confocal LSM (LSM510; Carl Zeiss GmbH) with a 63 × 1.4 DIC Plan-Apochromat oil-immersion objective. Z-stacks were acquired at an interval of 0.5 µm.

Co-localization events were analysed by assessing fluorescence intensity profiles in individual sections and Z-stacks projections were used for image presentations. Three-dimensional image models for fly-through movie presentations were generated from 0.2 µm Z-stack image files using Volocity 3.1 software (Improvision)[43]. The mode isosurface was used to identify a surface around the objects, where all voxel intensity values were the same.

For autophagy assessments fluorescent punctate structures were counted using at least 300 cells per condition from three independent experiments. Results were presented as the percentages of cells displaying punctate structures. The number of punctate structures per cell was quantified using confocal images and ImageJ (Analyse Particles).

**Automated high through-put fluorescence microscopy.** Automated image acquisition and analysis was conducted in 96-well cell culture plates. Cells, treated or not, were fixed in paraformaldehyde (3.7% in PBS), cell nuclei were stained with 4,6-diamidino-2-phenylindole (DAPI) (AppliChem; 28718-90-3) and if appropriate immunostaining was conducted. Images were automatically acquired using the In Cell Analyser 1,000 (GE Healthcare; Nikon Plan Fluor ELWD 40 × 0,6 objective) and analysed using the In Cell Analyser 1,000 Workstation 3.4 software. Per well, 25 images were acquired and fluorescence puncta automatically quantified using the dual area object assay (IN Cell Analyser Workstation 3.4). Thereby, DAPI-stained nuclei were detected with a top hat segmentation algorithm (minimum size of 100 µm$^2$ per nucleus) and GFP overall fluorescence was used to detect both the cell area and the puncta of each individual cell (multiscale top hat algorithms): the cell area was detected with a characteristic area of 1,500 µm$^2$ per cell; puncta (between 0.5 and 5 µm) were detected with an intensity threshold > 200 (puncta intensity over cytoplasmic intensity fluorescene threshold > 1.15) (ref. 36).

**Lentiviral-based shRNA screening.** MISSION LentiExpress Human Kinases (Sigma-Aldrich; SHX001) was used to identify human protein kinases that modulate the number of GFP-WIPI1 punctate structures. Monoclonal U2OS cells stably expressing GFP-WIPI1 were seeded in 96-well plates pre-containing the lentiviral particles and 96 h after infection, cells were washed twice with DMEM GlutaMAX. For elevating basal autophagy cells were incubated in DMEM GlutaMAX (without FCS) for 3 h, fixed in paraformaldehyde (3.7% in PBS) and stained with DAPI (AppliChem; 28718-90-3). Automated image acquisition was conducted by using the In Cell Analyser 1,000 (GE Healthcare; Nikon 40 × Planfluor objective) and GFP-WIPI1 punctate structures automatically analysed with the In Cell Analyser 1,000 Workstation 3.4 software[31,36]. Candidates were chosen for further analysis if their downregulation resulted in a decrease (< 20%) or an increase (> 45%) of GFP-WIPI1 puncta-positive cells with at least two different shRNA constructs targeting the same protein kinase. For a secondary screen corresponding parallel candidate samples from the MISSION LentiExpress Human Kinases shRNA library were used to isolate U2OS-GFP-WIPI1 cell lines stably expressing candidate shRNAs. Therefore, U2OS-GFP-WIPI1 cells were infected with a pool of four different lentiviruses for each candidate kinase, selected and assessed for GFP-WIPI1 punctate cells upon starvation in nutrient-free medium (EBSS). Based

on the $P$ values calculated by comparisons with non-targeting shRNA control cell lines candidate kinases were chosen for final assessments with transient siRNA transfections. A flow diagram of the screening procedure along with results are presented in Supplementary Fig. 8.

**Electron microscopy.** G361 cells were treated with EBSS for 3 h and fixed with paraformaldehyde (3.7% in PBS) for 30 min. The cells were scraped off, collected and further fixed in 2% glutaraldehyde and 0.5% osmium tetroxide in 0.1 × PBS, dehydrated with ethanol, and embedded in Epon. Thin sections (ultramicrotome cuts) were contrasted with uranyl acetate and lead citrate and finally examined in an EM410 electron microscope (Philipis)[84,85]. Acquired EM section images (documented digitally using Ditabis) were used for estimating differences in the appearance of autophagosomal structures and precursors between control (shControl) and shWIPI cell lines.

**Long-lived protein degradation assay.** The proteolysis of long-lived proteins was assessed by culturing the cells with 0.1 μCi $^{14}$C-Valine (Hartmann Analytic; MC277) for 24 h, washed with PBS, and chased with cold valine (10 mM L-Valine; Sigma-Aldrich; 94,619) for 1 h to remove short-lived proteins[52]. Upon different treatments (for example, starvation) cells were lysed with RIPA buffer supplemented with SDS and proteins precipitated with ice-cold 10% trichloroacetic acid/1% phosphotungstic acid. Both, the soluble and the insoluble radioactivity was measured. Proteolysis was calculated as the percentage of TCA-soluble radioactivity in relation to the insoluble radioactivity.

**Phospholipid-protein overlay assay.** Native cell extracts (750 mM aminocraproic acid, 50 mM Bis-Tris, 0.5 mM EDTA, pH 7.0, Protease Inhibitor Cocktail) from G361 cells or U2OS cells stably expressing GFP-WIPI proteins were obtained and used to overlay membrane-immobilized phospholipid membranes (Echelon; P-6001) for standard ECL (WBKLS0500) detection of bound WIPI proteins (primary antibodies: anti-WIPI1, anti-WIPI2 or anti-WIPI4 (endogenous WIPI proteins); anti-GFP (GFP-tagged WIPI proteins)[38].

**MS and analysis.** Monoclonal U2OS cell lines stably expressing either of the GFP-tagged WIPI proteins or GFP alone were subjected to anti-GFP immuno-precipitations and purified protein complexes separated in a 12% NUPAGE Novex Bis-Tris Gel (Invitrogen) in MOPS SDS running buffer (Invitrogen) for 10 min at 200 V. The NUPAGE gel was stained with Colloidal Blue Staining Kit (Invitrogen), gel pieces were cut and digested in gel with trypsin as described previously. Peptide mixtures were desalted using C18 StageTips and separated on the EasyLC nano-high-performance liquid chromatography HPLC (Proxeon Biosystems) coupled to an LTQ Orbitrap XL (Thermo Fisher Scientific). Binding and chromatographic separation of the peptides was performed on a 15 cm fused silica emitter with an inner diameter of 75 μm (Proxeon Biosystems), in-house packed with reversed-phase ReproSil-Pur C18-AQ 3 μm resin (Dr Maisch GmbH). The peptide mixtures were injected in HPLC solvent A (0.5% acetic acid) at a flow rate of 500 nl min$^{-1}$ and subsequently eluted with an 127 min segmented gradient of 5-33-50-90% of HPLC solvent B (80% acetonitrile in 0.5% acetic acid) at a flow rate of 200 nl min$^{-1}$. The 10 most intense precursor ions were sequentially fragmented in each scan cycle using collision-induced dissociation. In all measurements, sequenced precursor masses were excluded from further selection for 90 s. The target values were 5,000 charges for MS/MS fragmentation and $10^6$ charges for the MS scan. Acquired MS spectra were processed with MaxQuant software package versions 1.2.2.9 with integrated Andromeda search engine. Database search was performed against a target-decoy *H. sapiens* database obtained from Uniprot (downloaded 25 February 2014), containing 88,692 protein entries, and 245 commonly observed contaminants. Trypsin was defined as the protease with a maximum missed cleavage of two. Oxidation of methionines, N-terminal acet-ylation, and phosphorylation of serine, threonine and tyrosine were specified as variable modifications, whereas carbamidomethylation on cysteines was defined as a fixed modification. Initial maximum allowed mass tolerance was set to 6 ppm (for the survey scan) and 0.5 Da for collision-induced dissociation fragment ions. A false discovery rate of 1% was applied at the peptide and protein level[57,58,86–89]. The MS proteomics data have been deposited to the ProteomeXchange Consortium via the PRIDE[90] partner repository with the dataset identifier PXD006119.

Selecting final candidates for this study here (see Supplementary Data 2), the following criteria were applied: (1) nuclear proteins (Uniprot gene ontology) identified were excluded, (2) proteins identified in GFP control complexes were excluded unless the peptide count of proteins identified in GFP-WIPI immunocomplexes divided by the peptide counts of proteins of the GFP control samples was higher than 10, (3) the sum of all peptide counts in independent sets was higher than 2, (4) peptides were identified in at least two independent experiments. For the identification of WIPI3-interacting proteins only the proteins with peptide counts present in the GFP control samples were excluded. GeneMANIA was used to visualize interactions identified in our study and to distinguish from previously reported or predicted interactions (network category 'physical interactions', weighting over 10).

**Subcellular fractionation.** A crude lysosomal fraction (CLF) was prepared from $2 \times 10^8$ U2OS-GFP-WIPI3 cells cultured for 3 h in DMEM GlutaMAX/10% FCS supplemented with 200 nM Bafilomycin A1. The cells were homogenized with a Dounce homogenizer (Pestle B), and two subsequent centrifugation steps at 1,000 g for 10 min and for 20 min at 20,000 g. Further purification of CLF was conducted by using the Lysosomal Isolation Kit (LYSISO1; Sigma-Aldrich). Briefly, extraction buffer was added to the pellet containing the CLF. To enrich lysosomes option A and B were used according to the manufacturer's protocol. Fractions of 500 μl were collected from the top of the gradient and used for immunoblot analysis.

**Structural homology modelling and sequence alignment.** Structural homology models for WIPI1, WIPI2B, WIPI3 and WIPI4 were generated using the HHpred web-service[32,41] and selecting the PDB structure 3vu4 (*Kluyveromyces marxianus*) as model reference. The multiple WIPI protein sequence alignment was generated by combining the pairwise alignments generated by HHpred to the reference structure 3vu4. Inconsistent gapping between the pairwise alignments and 3vu4 at the end of structural elements or within loops were manually resolved using the program AlnEdit (Java-based alignment editor at Konstanz University Applied Bioinformatics Lab http://bioinformatics.uni-konstanz.de/programs/alnedit/).

**Statistical analysis.** Statistical analysis of the data was performed by two-tailed heteroscedastic $t$-testing. $P$ values $<0.05$ were taken to be statistically significant (*$P<0.05$; **$P<0.01$, ***$P<0.001$).

**Data availability.** Source data for Figures and Supplementary Figs are provided as Supplementary Data Files with the article. Other data that support the findings of this study are available via GenBank (https://www.ncbi.nlm.nih.gov/genbank/) with identifier KX434429, and via ProteomeXchange (http://www.proteomexchange.-org/) with identifier PXD006119. All other relevant data supporting the findings of this study are available on request.

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

## Acknowledgements

We thank Noboru Mizushima, Tokyo Medical and Dental University, Japan for providing m5-7 cells, and Tamotsu Yoshimori, Osaka University, Japan for sharing the GFP-LC3 plasmid. We appreciate initial data collection by previous (Irina Kremenetskaia, Anke Jacob, Mario Mauthe. Simon G. Pfisterer, Katharina Sporbeck) coworkers (T.P.-C. laboratory). T.P.-C. acknowledges grant support from the German Research Foundation (DFG) (SFB 773, GRK 1302) and M.P.T. from the Swiss National Science Foundation (31003A_143739). Further financial support was provided by the Forschungsschwerpunktprogramm Baden-Wuerttemberg (Kapitel 1403, Tit. Gr. 74), Landesgraduiertenförderung (LGFG) Baden-Wuerttemberg and the International Max Planck Research School 'From Molecules to Organisms', Tuebingen, Germany.

## Author contributions

D.B., A.J.M., Z.T., A.-K.T., T.Z., D.Br., M.P.T., H.R. and T.P.-C. conducted experiments and analysed data. T.F. conducted multiple sequence alignments and structural homology modelling, M.F.-W. and B.M. performed NanoLC-MS/MS analysis. D.B., A.J.M., T.Z., Z.T. contributed to manuscript preparation. T.P-.C conceived the study and wrote the manuscript. All authors read and approved the final version of the manuscript.

## Additional information

**Competing interests:** The authors declare no competing financial interests.

