## [Peer Review File · Nature Communications]

Reviewers' Comments:

Reviewer #1 (Remarks to the Author)

Bakula and colleagues have addressed the question of the function of the four WIPI proteins using both a global and targeted set of experiments. This is an important and interesting question as in mammalian cells there is data to support the role of WIPI1 and WIPI2 but relatively little about WIPI4 and nothing about WIPI3.

The authors first assess the role of all 4 WIPIs in regard to PI3P binding and localization to early markers, showing the common features of all 4 proteins. They then continue on to more directed experiments to confirm and determine the role of the WIPIs in autophagy. To gain a broader overview of their function, and role in autophagy, they performed an interactome study using mass spec techniques. They show a number of new interactions both unique to each of the four WIPI proteins, and also interactors which connect the WIPI proteins. Interestingly they also show that WIPI1, 2 and 4 interact while WIPI3 does not interact with the others. The major emphasis of the work is on the network shown in Figure 3b, in particular the NUDC, AMPK, TSC1, TSC2. Messages.

Major comments

1. Overall, the LC3 lipidation data is very poor and there is missing in some cases BafA treatment, and or quantification. In particular, Fig. 2e, S2b, c, d, e. Some of this data has been previously published and done here poorly so it does not add anything new.
2. The EM is a valuable approach and reveals potentially important information. However, the phenotypes observed must be supported by quantification. For example, the rough ER tubular structures shown in Fig. 2b, and Fig. 2f which occur after WIPI2 and WIPI4 knockdown. This morphological data should be confirmed by morphometric analysis in particular if the authors want to conclude any similarities or difference between WIPI2 and WIPI4.
3. Page 6, final paragraph. The data on the melanosome should be removed as it is not an essential part of the manuscript and again would require quantification to be included.
4. Page 6, 3rd paragraph. The authors conclude that all WIPIs were able to associate with each other, but on Figure S3 it looks like myc-WIPI1 does not co-immunoprecipitate GFP-WIPI1.
5. Page 7, the authors state WIPI1 likely supports WIPI2 in Atg16L1 recruitment but present no mechanism. Furthermore, if WIPI1 supports WIPI2 it is difficult to understand why WIPI2 puncta are not decreased by WIPI1 knockdown (line 205).
6. Page 8, the authors show GFP-WIPI3 pulls down TSC1-TSC2 but they should also show endogenous TSC1-TSC2 can pulldown GFP-WIPI3 and not the other GFP-WIPI protein.
7. Figure S5a, b, and c it is very difficult to see the spots.
8. Page 8, line 250, the authors say they expect TSC2 knockdown to reduce GFP-WIPI3, WIPI1, and LC3 puncta. It is not clear why this is expected especially if TSC1 is still present. In addition, the authors should control for mTORC1 inactivation.
9. Page 9, the authors show data that suggests that GFP-WIPI3 interacts with and colocalizes with FIP200 on late endosomes and lysosomes upon starvation. Are these autolysosomes? Can the authors detect FIP200, TSC1 and GFP-WIPI3 under fed conditions? Does the interaction change with starvation?
10. The authors mutate conserved residues in WIPI4 but give no information about the basis for the conservation. Are these the only candidate conserved residues in WIPI4? How were they chosen?
11. Figure 7f is confusing as there is still AMPK in the GFP-WIPI2 pulldown. Figure 7g needs a GFP-only control and to be quantified, done as n=3 in duplicate.
12. The Discussion over states the data as the authors say they provide evidence that "all human WIPI proteins function...upstream of PI3P production". They have not provided definitive evidence for this statement.

Minor

1. In Fig. 4a, the input with GFP-WIPI2B is very low, and the experiment should be repeated.

Reviewer #2 (Remarks to the Author)

Summary

The manuscript "WIPI3 and WIPI4 β -propellers act as scaffolds for LKB1-AMPK and TSC1-TSC2 in the control of autophagy" from Bakula et al. provides compelling evidence for autophagy regulation by WIPI proteins. Importantly, with the exception of WIPI2, function and regulation of the different WIPI proteins remain largely elusive. The authors start out with a comprehensive and systematic analysis of the four different WIPI proteins with regard to their PtdIns3P binding properties and their requirement for autophagosome formation. Subsequently, the authors employed mass spectrometry-based proteomics to map the interactomes of all four WIPI variants. Among the binding candidates the authors selected NudC, AMPK, ATG16L1, TSC1, FIP200 and ATG2A for rigorous interaction validation and extensive functional characterization. Importantly, the latter set of experiments convincingly revealed the importance of these interactions for autophagy regulation. In addition, by conducting a shRNA-based human kinome screen the authors uncovered that the interaction between WIPI4 and ATG2A is regulated by LKB1/AMPK. Based on these results, Bakula and colleagues postulate that WIPI proteins regulate autophagy at several different levels.

The manuscript is elegantly written and all experiments are well conceived, thoroughly conducted and rigorously controlled. Moreover, many experiments are confirmed with different methods (e.g. IF and WB, or AMPK activation with compounds and glucose starvation), great detail (e.g. mapping the amino acids important for ATG2 binding on WIPI4) and statistical power (high throughput analysis). Overall, the amount of data is quite remarkably and it is a pity that some of this is only mentioned in one sentence (for example NudC as negative autophagy regulator (Supplement Fig 4)) and or summarized in one figure (for example the kinome screen (Fig 8a/Supplement Figure 8)). In my opinion this manuscript is suitable for publication in Nature Communications provided that the authors address only a few minor comments.

Minor points

- 1) Title and line 48/68... : Please write TSC complex or name all known TSC subunits TSC1-TSC2-TBC1D7 .
- 2) Fig 1d: Although it may be a bit space consuming, the authors should provide single channel pictures at least for the inlays.
- 3) Fig 2a: How specific are the shRNAs for the different WIPIs? The authors should cross check specificity of WIPI1-4 shRNAs using all WIPI primers in qPCR or alternatively perform WB with the endogenous antibodies available?
- 4) Line 267: Supplement figure 5a should be 6a.
- 5) Fig 7e: The authors should make the difference between the first and the second graph more obvious. For example by using "% of GFP-WIPI1-puncta-positive cells" as y-axis label as it is described in the figure legend.
- 6) Line 303/305: Fig 7f should be Fig 7g.

Reviewer #3 (Remarks to the Author)

This manuscript makes a number of interesting observations about the role of WIPI proteins and how they coordinate with AMPK and ULK1 to regulate discrete steps of autophagy. According to the title, the paper focuses on novel roles of WIPI3 and 4 as scaffolds for the TSC2 and AMPK signaling axis respectively in autophagy, but the paper is lacking some mechanism of the how these proteins actually contribute the control of autophagy. While general well controlled and

interpreted, some additional experiments or at a minimum clarification of the points raised below in the text would help to make these sections of the paper stronger.

1. The authors demonstrate nicely that WIPI3 the co-localization and association of WIPI3 with TSC1-TSC2 (and AMPK phosphorylated TSC2) and Lamp 2+ lysosomal structures upon starvation conditions, but what is lacking is the functional relevance of this association on autophagy. They postulate that WIPI3 contributes to TSC2 modulation of TORC1 signaling during starvation but don't show this association is important for TSC2 activity against TORC1 or localization of TSC2 to lysosomes. They do show that siTSC2 decreases WIPI3 puncta, but this decrease isn't surprising since TSC2 is not longer inhibiting mTOR (so less autophagy) which would decrease all autophagosome formation and therefore there will be less puncta of any autophagy marker and therefore does not prove a role specifically in regulating WIPI3.

2. If WIPI3 assists TSC2 with inhibition of TORC1, then is there evidence of enhanced TORC1 activity in the shWIPI3 cells? Is there less TSC2 at the lysosomes without WIPI3 or more active TORC1 at the lysosomes?

3. Would you expect a change in association of WIPI3 with the TSC complex in response to stimulation of autophagy by starvation if WIPI3-TSC2 interaction is important for autophagy? The text suggests from Figure 5B and C that more TSC2 pulls down in under starvation conditions, but it is not clear from the MEF blots (figure 5b) since the WIPI3 levels IP'ed are not equal and increase from fed to starved is very minor in figure 5C.

4. The authors demonstrate that WIPI3 associated with endogenous FIP200 and further colocalizes with AMPK during starvation, which is abolished after stimulation with amino acids. FIP200 knockdown inhibits puncta formation in WIPI3 or WIPI1 expressing cells. This is not surprising given that FIP200 is crucial for initiation of autophagy, it doesn't really speak to the importance of its interaction with WIPI3. Does knockdown of WIPI3 change FIP200 localization, association with ULK1, and affect FIP200 function in autophagy?

5. The authors identified critical residues in WIPI4 that mediate its ability to bind efficiently to ATG2A (N15A and D17A) and knockdown of these two gene causes increase in the size and number of puncta, so they conclude WIPI4-ATG2 complex regulates the size of autophagosomes. It would be very interesting to employ the non-binding mutants and look at the effects on autophagosome size. How does this complex control size of autophagosomes? Also of note, these residues are completely conserved across the WIPI proteins, so why the specificity for WIPI4?

6. The authors demonstrate that endogenous AMPK and ULK1 associate with WIPI4-ATG2A more in fed conditions than nutrient or glucose starvation. What mediates this change in interaction? Does the LKB1 phosphorylation site on AMPK influence complex formation ie AMPK activation? Does kinase dead AMPK associate with the complex equally in starved conditions or alternatively no longer associate with WIPI4 in fed conditions? Does kinase dead ULK1 associate with WIPI4? Can either of these kinases phosphorylate WIPI4 or ATG2A?

7. How does binding to or release from AMPK-ULK1 modulate autophagosome size? Can authors speculate on this in discussion at least?

8. The authors performed a shRNA kinase library screen to identify kinases that effect WIPI1 puncta formation in starved conditions with a secondary screen of amino acid starvation. BRSK and NUAK were identified (although barely significant p values) and knockdown of these and AMPK decrease WIPI4 puncta slightly so they suggest AMPKs regulate WIPI4 . First, it is interesting that ULK1 was not pulled out of this screen, as it is a known regulator of autophagy that they showed can bind WIPI4. Do NUAK or BRSK bind to WIPI4? Does knockdown of NUAK or BRSK phenocopy WIPI4 knockdown at all?

9. Some of the data presented (Supplementary Figure 4 on NudC and Figure 4 on WIPI1 and 2) could be excluded for the sake of clarity of the story. Both figures speak to potential interactors of WIP proteins in the autophagy pathway, but with the focus of the paper primarily on WIP3 and 4 interactions with TORC1 and AMPK pathways (as indicated by the title and subsequent figures) these data are unnecessary. The NudC data is the more interesting and novel of these 2 pieces of data, so would be the more interesting one to keep in the main paper if the paper is framed more as a full WIPI-autophagy-interactome paper. A section of supplementary information is dedicated specifically to go through detailed analysis of this figure, which seems like it should either be included as its own section in the manuscript or be omitted. If it is included some additional questions need commenting on by authors if not experimentally addressed:

A) Is NudC in a complex with all three WIPI (1,2,4) together or are their separate subsets of interactions that could potentially have different functions?

B) Do these proteins associate through their beta-propellers?

C) It would be nice to have quantitation of the increase in autophagosomes upon NudC knockdown.

D) If this complex negatively regulates autophagy, then one could imagine that NudC should dissociate from the WIPIs under starvation conditions to relieve this repression, but that is not the case. How then do they become active (new binding partners, changes in localization perhaps?).

Minor points

1. Figure 1 title should include "under starvation"

2. Figure 1b left and right panels should be separated since they are different experiments in different cells, which is a bit confusing. Also would be worth swapping the data using endogenous WIPIs from G361 cells to supplement and move all the overexpressed U2OS data to the main figure so 1B doesn't contain two different cell lines. Ideally it would be nice to western data to look at the relative expression of the endogenous proteins in G361 and U2OS cells the supplement as well.

3. Consistent labeling of WIPI2B and WIPI2D throughout the figures would be helpful (instead of sometimes labeling just WIPI2).

4. For Figure 2b, it would be helpful to have quantitation of number of normal and abnormal autophagosomes in the different lines.

5. Supplementary figure 2d, the way the Western is cropped, it looks like a shift of LC3 type but it is actually accumulation of the lipidated LC3 as stated in the text.

6. In Figure 5f, it would be helpful to have quantification of colocalization and to see if there is colocalization in fed cells as well.

7. Figure 5I, it is obvious that the re-feed is not activating TORC1 very well. Western is not very convincing

8. In Figure 8b, why is the knockdown assessed in the WIPI1-expressing U2OS line when the functional experiment (8c) is in the WIPI4 line (or is this a typo)?

9. In multiple places in the paper, it is mentioned that WIPI3 and 4 have functions downstream of LC3 but in the model (8g) all the roles are depicted upstream or converging on LC3.

Reviewer #4 (Remarks to the Author)

In this report by Proikas-Cezanne and co-workers, the authors describe a scaffolding function for WIPI3 and WIPI4 proteins to link LKB1-AMPK and TSC1-TSC2 signaling in the control of autophagy.

This is a well-conceived and well-designed study that leverages a kinome-wide screen with

imaging and proteomics to identify factors involved in autophagosome formation. The functional proteomics data in the manuscript appear to be of high quality, and the mass spectrometry methods and approaches are well described.

Some additional discussion or experimentation would be helpful to further clarify or substantiate the claims of distinct protein complexes and linkages between protein interactors, rather than just relying on peptide counting, for example, "as we found only a few peptide counts for ATG16L in our WIPI1 MS analysis, we suggest that the interaction between WIPI1 and ATG16L occurs indirectly via WIPI2." It would be more convincing to perhaps knock down WIPI2 and demonstrate loss of ATG16L interaction by either mass spectrometry or western blotting.

Otherwise, I have no reservations about the quality of the mass spectrometry-based data in the manuscript.

Reviewer #1 (Remarks to the Author):

Bakula and colleagues have addressed the question of the function of the four WIPI proteins using both a global and targeted set of experiments. This is an important and interesting question as in mammalian cells there is data to support the role of WIPI1 and WIPI2 but relatively little about WIPI4 and nothing about WIPI3.

The authors first assess the role of all 4 WIPIs in regard to PI3P binding and localization to early markers, showing the common features of all 4 proteins. They then continue on to more directed experiments to confirm and determine the role of the WIPIs in autophagy. To gain a broader overview of their function, and role in autophagy, they performed an interactome study using mass spec techniques. They show a number of new interactions both those unique to each of the four WIPI proteins, and also interactors which connect the WIPI proteins. Interestingly they also show that WIPI1, 2 and 4 interact while WIPI3 does not interact with the others. The

major emphasis of the work is on the network shown in Figure 3b, in particular the NUDC, AMPK, TSC1, TSC2.

We deeply appreciate the thorough review and thank the reviewer very much for his/her positive words. We are most grateful for the reviewers' constructive suggestions, which prompted us to amend our study on the basis of the reviewer's advice offered.

Major comments

1. Overall, the LC3 lipidation data is very poor and there is missing in some cases BafA treatment, and or quantification. In particular, Fig. 2e, S2b, c, d, e. Some of this data has been previously published and done here poorly so it does not add anything new.

- We are most grateful for the reviewer's salient observation, which guided us to revisit the LC3 lipidation data provided in our original manuscript. We completely agree that the LC3 lipidation data for both WIPI1 and WIPI2 knockdown (KD) did not add anything new with regard to previously published data. Accordingly, we removed this data but kept the data on significantly reduced long-lived protein degradation upon starvation in WIPI1 KD cells, as this provides a new information (Suppl. Fig. 2g in the revised manuscript).
- As follows, we restructured Fig. 2 to focus more on WIPI3 KD and WIPI4 KD assessments. We have retained the original Fig. 2d showing that in both WIPI3 KD and WIPI4 KD conditions the number of LC3 puncta/cell significantly increased. Subsequently, we added our data on the significant increase of endogenous WIPI2 puncta upon WIPI3 KD and WIPI4 KD as new Fig. 2e (former Suppl. Fig. 2f). Combined the results suggest that the significant increase of both LC3 puncta and WIPI2 puncta in WIPI3 KD and WIPI4 KD cells are due to a block downstream of LC3. This result is in agreement with the study by Dooley et al. demonstrating that WIPI2 recruits the ATG16L complex for LC3 lipidation (Dooley et al., Mol Cell 2014; Ref. 26). With regard to WIPI4, this result is also in line with (i) the finding that WIPI4 mutations, conferring a loss of WIPI4 function and identified in SENDA patients, block autophagy downstream of LC3 (Saitou et al., Nat Genet. 2013), and (ii) data by Lu et al. on EPG-6, the WIPI3/4 homologue in *C. elegans*, suggesting that EPG-6 functions downstream of LC3 (Lu et al., Dev Cell 2010).
- We recognized that it would indeed improve the revised version of the manuscript if LC3 lipidation assessments in WIPI3 KD and WIPI4 KD settings would be quantified. Therefore, we repeated this analysis in both fed and starved conditions (in presence or absence of bafilomycin A1) and quantified the western blot results (Fig. 2f, Suppl. Fig. 2h).
- We found that in fed conditions, LC3-II significantly accumulated in WIPI3 KD and WIPI4 KD settings (Fig. 2f, Suppl. Fig. 2h). With regard to WIPI4, this result is in agreement with the LC3 lipidation analysis of SENDA patients harboring WIPI4 mutations that indeed show an increase in LC3-II abundance when compared to healthy controls (Saitou et al., Nat Genet. 2013). Here, we additionally observed that LC3-I also significantly increased in fed conditions in WIPI4 KD, and also in WIPI3 KD settings. Hence our new assessment further underlines that WIPI4, and also WIPI3, should play a role downstream of LC3 during the process of autophagy.
- Further, our new LC3 lipidation assessment in starved conditions (Fig. 2f, Suppl. Fig. 2h) showed that in both WIPI3 KD and WIPI4 KD settings the abundance of LC3-I and LC3-II did not significantly change when compared to control cells. We suggest that this may imply a layer of functional

redundancy of WIPI proteins at the nascent autophagosome during starvation-induced autophagy. In line with this observation, SENDA patients harboring WIPI4 mutations (Saitou et al., Nat Genet. 2013) are viable, despite the severe neurological phenotype. Hence although WIPI4 is mutated and blocks autophagy downstream of LC3 in this disease context autophagy is not completely abolished.

- The above results, former and new, are summarized in the revised result section as follows (revised sections in yellow):

“...we anticipated that lipidated LC3 would accumulate in the absence of WIPI3 or WIPI4, which was indeed the case when we looked for GFP-LC3 puncta using automated high-throughput imaging⁴² (Fig. 2d). In line with this observation, we also found that endogenous WIPI2 puncta accumulated in WIPI3 KD and WIPI4 KD cells due to the downstream blockade of LC3 (Fig. 2e). Moreover, by western blotting we observed a significant increase in basal levels of lipidated (LC3-II) and unlipidated LC3 (LC3-I) (Fig. 2f, Suppl. Fig. 2h). Upon amino acid starvation, however, LC3 levels did not significantly change (Suppl. Fig. 2h), indicating a layer of redundancy between the different WIPI proteins.”

2. The EM is a valuable approach and reveals potentially important information. However, the phenotypes observed must be supported by quantification. For example, the rough ER tubular structures shown in Fig. 2b, and Fig. 2f which occur after WIPI2 and WIPI4 knockdown. This morphological data should be confirmed by morphometric analysis in particular if the authors want to conclude any similarities or difference between WIPI2 and WIPI4.

- We absolutely agree with the reviewer's important point. Accordingly, we now provide the requested quantification in the revised version of our manuscript (raw data in Suppl. Table 1, excel sheet labelled “Suppl. Fig. 2”). Based on this analysis we restructured Suppl. Fig. 2 to represent the structures quantified morphometrically in each of the KD setting for WIPI1-4 (Suppl. Fig. 2b-2f).
- Our estimation shows that rough ER tubular structures increased by 4.93 fold in WIPI2 KD and 3.91 fold in WIPI4 KD settings. Exemplary, these rough ER tubular structures are now shown magnified for both WIPI2 KD (Suppl. Fig. 2d) and WIPI4 KD (Suppl. Fig. 2f) cells.
- Moreover, counting autophagosomal structures in EM sections we also provide new results that highlight a reduction of autophagosomal structures in all WIPI KD settings: reduction by 2.65 fold in WIPI1 KD, 2.94 fold in WIPI2 KD, 6.42 fold in WIPI3 KD and 3.47 fold in WIPI4 KD (Suppl. Table 1).
- In addition, the appearance of cup-shaped double membrane structures, resembling elongated phagophores in WIPI3 KD and WIPI4 KD cells have also been quantified and more examples are shown in the Suppl. Fig. 2e (WIPI3 KD) and Suppl. Fig. 2f (WIPI4 KD).
- The revised version of our manuscript now includes this new assessment in the Suppl. Table 1 (excel sheet labelled “Suppl. Fig. 2”) and Suppl. Fig. 2b-f. The manuscript text of the section titled “Functional WIPI3 and WIPI4 knockdown impairs appropriate autophagosome formation” is accordingly changed (changes are highlighted in yellow in the manuscript text file) and the corresponding passage now reads as follows:

“Compared with the control cell line (shControl, Fig. 2b, Suppl. Fig. 2b), WIPI1 KD cells did produce autophagosomes upon starvation (Fig. 2b, panel shWIPI1; Suppl. Fig. 2c), although we observed a 2.65-fold reduced presence of autophagosomal structures (Suppl. Table 1). In line with this finding, autophagic flux⁵⁰ assessments by LC3 lipidation analysis was previously reported to be negatively affected in WIPI1 KD cells²⁴. Additionally, we here found that starvation-induced degradation of long-lived proteins³² was significantly reduced in WIPI1 KD cells (Suppl. Fig. 2g). EM of WIPI2 KD revealed that proper autophagosome formation was negatively affected as inferred from a 4.93-fold accumulation of rough endoplasmic reticulum (RER) tubular structures (Fig. 2b, panel shWIPI2; Suppl. Fig. 2d), prominently marking an early blockade of autophagosome formation²⁶ prior to phagophore formation (template membranes from which autophagosomes emerge^{23, 53}) (Suppl. Table 1). In line with this finding, the appearance of autophagosomal structures decreased by 2.94 fold when compared to control cells (Suppl. Table 1). Moreover, in WIPI2 KD cells, the number of GFP-LC3 puncta was significantly reduced (Fig. 2c), corroborating the previous finding that WIPI2 is required for LC3 lipidation^{25, 26}. The above results confirm that both WIPI1 and WIPI2 function upstream of LC3, as previously suggested³⁵. In addition, the results

underline the conception that WIPI2 is required for autophagosome formation, whereas WIPI1 is dispensable to a certain degree.

Interestingly, both WIPI3 KD (Fig. 2b, panel shWIPI3; **Suppl. Fig. 2e**) and WIPI4 KD (Fig. 2b, panel shWIPI4; **Suppl. Fig. 2f**) resulted in the appearance of cup-shaped double-membrane structures resembling elongated phagophore formation sites (Fig. 2b, **Suppl. Fig. 2e, f**, **Suppl. Table 1**). In both, **WIPI3 KD and WIPI4 KD cells the formation of autophagosomal structures decreased respectively by 6.42 and 3.47 fold when compared with shControl cells (Suppl. Table 1)**. In addition, RER tubular structures accumulated in the absence of WIPI4, as was observed in WIPI2 KD cells (**Suppl. Fig. 2f, Suppl. Table 1**).

3. Page 6, final paragraph. The data on the melanosome should be removed as it is not an essential part of the manuscript and again would require quantification to be included.

- We thank the reviewer very much indeed for this pivotal suggestion, which we immediately followed. We removed the data on the appearance of melanosomes from Fig. 2. In the manuscript text, however, we kept this as a minor note as this feature is very prominent. We feel that we cannot completely subtract this observation as this has also been observed before in WIPI1 KD (Ho et al., JBC 2011) and ULK1 KD (Kalie et al., PLoS one 2013) settings.
- In the revised version of the manuscript the corresponding section now reads as follows:

"Of additional note, EM observation of WIPI KD in the G361 melanoma cell line showed that **WIPI KD cells** were filled with melanosomes that were not detected in control cells (**data not shown**). The appearance of melanosomes upon ULK1 and WIPI1 silencing has been reported^{54,55}. Together with our **observation, this suggests** that autophagosome and melanosome formation **may be** co-regulated⁵⁶.

4. Page 6, 3rd paragraph. The authors conclude that all WIPIs were able to associate with each other, but on Figure S3 it looks like myc-WIPI1 does not co-immunoprecipitate GFP-WIPI1.

- We thank the reviewer very much for bringing this important point to our attention. We fully agree that the co-immunopurification provided in our original manuscript did not prominently demonstrate that GFP-WIPI1 associated with myc-tagged WIPI1. We now include additional data in the new Suppl Fig. 3b, clearly demonstrating that GFP-WIPI1 pulls down myc-tagged WIPI1.

5. Page 7, the authors state WIPI1 likely supports WIPI2 in Atg16L1 recruitment but present no mechanism. Furthermore, if WIPI1 supports WIPI2 it is difficult to understand why WIPI2 puncta are not decreased by WIPI1 knockdown (line 205).

- We very much like to thank the reviewer for this insightful comment. We revisited and discussed plausible interpretations of the results obtained and argue for a WIPI2-supporting role for WIPI1 in the recruitment of ATG16L1. This argument is based on the finding that significantly less endogenous ATG16L1 puncta were observed in WIPI1 KD settings (Fig. 4c). If WIPI1 were not to support the recruitment of the ATG16L complex by WIPI2, a transient knockdown of WIPI1 should not affect ATG16L puncta formation.
- We assume that WIPI2 may be recruited before WIPI1 binds PI3P at the nascent autophagosome, and that this specific WIPI2 localization is required for subsequent WIPI1 binding. If this were the case, one would (i) not observe a difference in WIPI2 puncta in WIPI1 KD settings, and (ii) not find less WIPI1 puncta in WIPI2 KD settings: both predictions are met by the data of our study.
- Nevertheless, the above reasoning represents plausible arguments, while the mechanism is not yet known and has to be investigated in the future.

6. Page 8, the authors show GFP-WIPI3 pulls down TSC1-TSC2 but they should also show endogenous TSC1-TSC2 can pull down GFP-WIPI3 and not the other GFP-WIPI protein.

- We are truly thankful for the reviewer's exceptional suggestion, which led us to intensively work on this issue by acquiring and establishing new reagents (antibodies and more sensitive ECL detection systems). Successfully, we were able to demonstrate that endogenous TSC1 (anti-TSC1 IP) associates with endogenous TSC2 (anti-TSC2 IB), and with endogenous WIPI3 (anti-WIPI3 IP) but importantly, not with the other WIPI proteins. This new result is shown in the new Fig. 5b.

7. Figure S5a, b, and c it is very difficult to see the spots.

- We agree with the reviewer and accordingly, have restructured the Suppl. Fig. 5 to now include complete cell images in single channels, along with the corresponding magnified sections shown in the main Fig. 5.

8. Page 8, line 250, the authors say they expect TSC2 knockdown to reduce GFP-WIPI3, WIPI1, and LC3 puncta. It is not clear why this is expected especially if TSC1 is still present. In addition, the authors should control for mTORC1 inactivation.

- The reviewer is right that also TSC1 plays an important function in mTORC1 inactivation. Nevertheless, the GAP activity of TSC2 is critical for mTORC1 inhibition via regulating Rheb. Thus, the KD of TSC2 alone should be sufficient to induce autophagy or rather to abolish the function of the TSC1-TSC2 complex.
- As we do not focus on a detailed and comprehensive mTORC1 assessment in this manuscript, we have rephrased this section and removed the remark "as expected".
- Assessing mTORC1 activity in TSC complex/WIPI3 KD settings is in fact a new project in the laboratory. Our initial observations, however, argue that the responses are similar in TSC1 KD (we have started this new project with TSC1 assessments) and WIPI3 KD settings. We found that both, mTORC1-dependent ULK1 phosphorylation (P-S757) increased in the absence of TSC1 and also, mildly in the absence of WIPI3, and that additionally, ULK1 protein abundance significantly increased in both TSC1 KD and WIPI3 KD settings (see below). This initial result indicates, that consequences of TSC complex/WIPI3 deficiency is complex and affects e.g. ULK1 on both transcriptional and post-transcriptional levels. Based on this we feel more confident to provide a thorough mTORC1 assessment, along with cellular consequences, in a more detailed follow-up study in follow-up the future.

9. Page 9, the authors show data that suggests that GFP-WIPI3 interacts with and colocalizes with FIP200 on late endosomes and lysosomes upon starvation. Are these autolysosomes? Can the authors detect FIP200, TSC1 and GFP-WIPI3 under fed conditions? Does the interaction change with starvation?

- We thank the reviewer very much for this important question. Indeed, we find that the colocalization between WIPI3 and TSC1 as well as FIP200 increases upon starvation, as WIPI3 and FIP200 puncta are in very low abundance in fed conditions. Further, colocalisation between WIPI3 and LAMP2 can be detected in low abundance in fed conditions, however, this is more prominent upon the addition of bafilomycin A1, and is abolished upon the inhibition of PI3P production (Suppl. Fig. 5c). To express this observation we added the following to the revised version of our manuscript:

"In fed cells GFP-WIPI3 also colocalised with LAMP2, however, this colocalisation was more apparent upon lysosomal inhibition by bafilomycin A1 (Suppl. Fig. 5c, upper panel) and was sensitive to PI3K inhibition (Suppl. Fig. 5c, lower panel)."

- We intend to provide a thorough colocalisation assessment of the complex relationship between WIPI3, TSC1-TSC2, FIP200 and LAMP2 under various experimental conditions in a subsequent study, and we very much hope that the reviewer can agree that this complex analysis is beyond the scope of the current manuscript, which focusses on the newly identified interactions. We end our manuscript with the outlook on subsequent work as follows:

"Taken together, our data reveal the WIPI interactome and suggest that the LKB1-AMPK regulatory network directly or indirectly regulates the scaffold functions of the four human WIPI proteins. Our work provides a frame work for further studies concentrating in more detail on the various new aspects presented here."

10. The authors mutate conserved residues in WIPI4 but give no information about the basis for the conservation. Are these the only candidate conserved residues in WIPI4? How were they chosen?

- We apologize for not making this important point clear in our original version of the manuscript.
- Initially, we have identified homologous residues in WIPI proteins that are also conserved in all PROPPINs (Proikas-Cezanne et al., Oncogene 2004).
- In the revised version of our manuscript on page 10 we added the following changes:

"By mutating individual amino acids in WIPI4 that are conserved in human WIPI proteins³² (Suppl. Fig. 1a) we identified two critical residues in WIPI4, N15 and D17, that confer ATG2A binding (Fig. 7b, Suppl. Fig. 7a)."

Reference 32 refers to our original publication from 2004.

- In addition, the revised Suppl. Fig. 1a now highlights the conserved residues in WIPI proteins in the alignment. The corresponding Suppl. Fig. 1a figure legend now reads accordingly as follows:

"Multiple protein sequence alignments of WIPI1, WIPI2B, WIPI3 and WIPI4 are presented. Two arginine residues crucial for phospholipid binding are conserved and highlighted with bold red letters in all WIPI sequences. Further amino acids homologous in WIPI proteins and all further members of the PROPPIN family³² are highlighted with red letters in WIPI1 only. Black letters in the WIPI3 protein sequence represent the original sequence (referred to as WIPI3S hereafter), blue letters indicate the new extended WIPI3 N-terminal sequence cloned in this study (GenBank accession number KX434429)."

Reference 32 refers to our original publication from 2004.

11. Figure 7f is confusing as there is still AMPK in the GFP-WIPI2 pulldown. Figure 7g needs a GFP-only control and to be quantified, done as n=3 in duplicate.

- We very much agree with the reviewer on this remark and have accordingly added the following new data, described below, in the revised version of our manuscript.

- The new Fig. 7f shows more clearly, that AMPK is found to co-immunoprecipitate with GFP-WIPI4, and not GFP-WIPI2B. Additionally we included the required GFP only control.
- We included also new experimental data that also includes the GFP control (n=3, each in duplicates), and we show that significantly less AMPK is co-immunoprecipitated when AMPK is activated by AICAR (new Fig. 7h and 7i).
- Moreover, we found that the WIPI4 D113A mutant was impaired in AMPK binding (Suppl. Fig. 8c), and that the WIPI4 D113A mutant did not respond with an increase in puncta formation upon glucose starvation (new Fig. 8g) and that the number of puncta-positive cells were not reduced in AMPK KD settings (new figure 8h) when compared to wild-type GFP-WIPI4.

12. *The Discussion over states the data as the authors say they provide evidence that "all human WIPI proteins function...upstream of PI3P production". They have not provided definitive evidence for this statement.*

- We absolutely agree with the reviewer's judgement and have accordingly rephrased the statement as follows:

"Here, we provide evidence that all human WIPI proteins function as scaffold building units that interlink the control of autophagy with the formation of functional autophagosomes."

Minor

1. *In Fig. 4a, the input with GFP-WIPI2B is very low, and the experiment should be repeated.*

- We thank the reviewer very much for this vital remark. Accordingly, we repeated this experiment and show in the new Fig. 4a visible amounts of GFP-WIPI2B in the input.

Reviewer #2 (Remarks to the Author):

The manuscript "WIPI3 and WIPI4 β -propellers act as scaffolds for LKB1-AMPK and TSC1-TSC2 in the control of autophagy" from Bakula et al. provides compelling evidence for autophagy regulation by WIPI proteins. Importantly, with the exception of WIPI2, function and regulation of the different WIPI proteins remain largely elusive. The authors start out with a comprehensive and systematic analysis of the four different WIPI proteins with regard to their PtdIns3P binding properties and their requirement for autophagosome formation. Subsequently, the authors employed mass spectrometry-based proteomics to map the interactomes of all four WIPI variants. Among the binding candidates the authors selected NudC, AMPK, ATG16L1, TSC1, FIP200 and ATG2A for rigorous interaction validation and extensive functional characterization. Importantly, the latter set of experiments convincingly revealed the importance of these interactions for autophagy regulation. In addition, by conducting a shRNA-based human kinome screen the authors uncovered that the interaction between WIPI4 and ATG2A is regulated by LKB1/AMPK. Based on these results, Bakula and colleagues postulate that WIPI proteins regulate autophagy at several different levels.

The manuscript is elegantly written and all experiments are well conceived, thoroughly conducted and rigorously controlled. Moreover, many experiments are confirmed with different methods (e.g. IF and WB, or AMPK activation with compounds and glucose starvation), great detail (e.g. mapping the amino acids important for ATG2 binding on WIPI4) and statistical power (high throughput analysis). Overall, the amount of data is quite remarkably and it is a pity that some of this is only mentioned in one sentence (for example NudC as negative autophagy regulator (Supplement Fig 4)) and or summarized in one figure (for example the

kinome screen (Fig 8a/Supplement Figure 8)). In my opinion this manuscript is suitable for publication in Nature Communications provided that the authors address only a few minor comments.

We sincerely thank the reviewer very much for the exceptionally encouraging comments on the quality of our manuscript. We were extremely stimulated by this evaluation and conducted the experiments requested for the revision process with great joy and enthusiasm.

Minor points

1) *Title and line 48/68... : Please write TSC complex or name all known TSC subunits TSC1-TSC2-TBC1D7 .*

- We thank the reviewer very much for this correction and followed the advise given; hence we wrote “TSC complex” in both title and manuscript text (changes are highlighted in yellow in the revised version of our manuscript).

2) *Fig 1d: Although it may be a bit space consuming, the authors should provide single channel pictures at least for the inlays.*

- We are most grateful for this important point being brought to our attention. Accordingly, we included single channel pictures for the inlays in the new supplementary figure 1f and apologize for omitting these images initially. In addition, we provide single channel images in the new Suppl. Fig. 5 and 6.

3) *Fig 2a: How specific are the shRNAs for the different WIPIs? The authors should cross check specificity of WIPI1-4 shRNAs using all WIPI primers in qPCR or alternatively perform WB with the endogenous antibodies available?*

- We absolutely agree with the reviewer that this comparison needs to be shown. We now include the complete assessment of our stable shRNA cell lines in the new Suppl. Fig. 2a, demonstrating specificity of employed WIPI1-4 shRNAs.

4) *Line 267: Supplement figure 5a should be 6a.*

- We thank the reviewer very much for this true observation. We altered the labeling accordingly in the revised version of our manuscript.

5) *Fig 7e: The authors should make the difference between the first and the second graph more obvious. For example by using “% of GFP-WIPI1-puncta-positive cells” as y-axis label as it is described in the figure legend.*

- We are grateful for this notion and we absolutely agree that the former labeling did not highlight the different assessments. We have now labeled the y-axis as suggested with “Puncta-positive cells (%)”.

6) *Line 303/305: Fig 7f should be Fig 7g.*

- According to the suggestion of reviewer 1 we added new data to Fig. 7 that further support the association of WIPI4-ATG2A with AMPK. Thus, the restructured Fig. 7 now includes new data and the labeling was adjusted accordingly.

Reviewer #3 (Remarks to the Author):

This manuscript makes a number of interesting observations about the role of WIPI proteins and how they coordinate with AMPK and ULK1 to regulate discrete steps of autophagy. According to the title, the paper focuses on novel roles of WIPI3 and 4 as

scaffolds for the TSC2 and AMPK signaling axis respectively in autophagy, but the paper is lacking some mechanism of the how these proteins actually contribute the control of autophagy. While general well controlled and interpreted, some additional experiments or at a minimum clarification of the points raised below in the text would help to make these sections of the paper stronger.

We are deeply grateful to the reviewer for the insightful comments and suggestions offered and we highly appreciate the important questions raised regarding our findings. Many of the points addressed by the reviewer led us to introduce vital changes to the revised manuscript (see below). At the same time, we were unable to experimentally respond to all important points raised, as the current projects in our laboratory - sparked by the new insights presented in both original and revised versions of the manuscript - are multifariously and request more long-term assessments. Also, in the discussion section we intended to remain rather conservative with regard to speculations on functional consequences as “our study provides a frame work for further studies concentrating in more detail on the various new aspects presented here.” (final sentence in our discussion). In the following, we respond to the reviewer's comments by sharing preliminary results that are directed towards a furthergoing understanding of functions of WIPI proteins.

1. The authors demonstrate nicely that WIPI3 the co-localization and association of WIPI3 with TSC1-TSC2 (and AMPK phosphorylated TSC2) and Lamp 2+ lysosomal structures upon starvation conditions, but what is lacking is the functional relevance of this association on autophagy. They postulate that WIPI3 contributes to TSC2 modulation of TORC1 signaling during starvation but don't show this association is important for TSC2 activity against TORC1 or localization of TSC2 to lysosomes. They do show that siTSC2 decreases WIPI3 puncta, but this decrease isn't surprising since TSC2 is not longer inhibiting mTOR (so less autophagy) which would decrease all autophagosome formation and therefore there will be less puncta of any autophagy marker and therefore does not prove a role specifically in regulating WIPI3.

- We thank the reviewer very much indeed for his/her overall positive appraisal of our work. We wish to highlight that we here provide the assessment of WIPI3 on the basis of our proteome analysis that we verified for TSC1-TSC2 and FIP200 in this study. It may be anticipated that WIPI3 puncta decrease in the absence of TSC2, however, it has not been shown before that WIPI3 in fact plays a role in autophagy at all. Our manuscript now evidences the contribution of WIPI3 to the process of autophagy (e.g. Fig. 1). We also show that WIPI3 localises to both the nascent autophagosome and the LAMP2 compartment, along with the TSC complex, and also with FIP200. Likely, WIPI3 mediates the association between TSC1 and FIP200 as we found that WIPI3 co-immunoprecipitates with a TSC1 fragment previously shown to interact with FIP200 (Fig. 5e).
- We predict that the functional consequences of the WIPI3 interactions are complex, as we found that WIPI3 KD, as well as TSC1 KD increased the abundance of ULK1 protein (see above our discussion and results above, as provided in response to reviewer #1 on page 5). In the TSC1 KD setting we find an increase in TORC1-dependent ULK1 phosphorylation that inhibits autophagy. In WIPI3 KD settings this is also indicated - but prominent to both settings - the abundance of ULK1 protein significantly changed. From this, the consequences on the great variety of TORC1 activities need to be addressed in a subsequent, new project that includes assessments on both transcriptional and post-transcriptional levels.

2. *If WIPI3 assists TSC2 with inhibition of TORC1, then is there evidence of enhanced TORC1 activity in the shWIPI3 cells? Is there less TSC2 at the lysosomes without WIPI3 or more active TORC1 at the lysosomes?*

- Along with our response to the first vital point of this reviewer, we have initiated such assessments and based on preliminary results in WIPI3 KD settings, mTOR positioning in the perinuclear region may indeed be affected (data not shown).
- Based on the elegant study by the Ktistakis laboratory (Manifa et al., eLIFE 2016), TORC1 positioning at the lysosome is extremely transient, hence this aspect will be part of our future investigations.

3. *Would you expect a change in association of WIPI3 with the TSC complex in response to stimulation of autophagy by starvation if WIPI3-TSC2 interaction is important for autophagy? The text suggests from Figure 5B and C that more TSC2 pulls down in under starvation conditions, but it is not clear from the MEF blots (figure 5b) since the WIPI3 levels IP'ed are not equal and increase from fed to starved is very minor in figure 5C.*

- We very much thank the reviewer for this insightful observation. Indeed, we find a more prominent colocalisation between WIPI3 and the TSC complex in starved conditions. At the same time, we predict that a future more detailed analysis of WIPI3 interactions in different fed and starved cellular settings will provide us with a particular experimental condition to specifically address TSC2 GAP activity and TORC1 activities.

4. *The authors demonstrate that WIPI3 associated with endogenous FIP200 and further colocalizes with AMPK during starvation, which is abolished after stimulation with amino acids. FIP200 knockdown inhibits puncta formation in WIPI3 or WIPI1 expressing cells. This is not surprising given that FIP200 is crucial for initiation of autophagy, it doesn't really speak to the importance of its interaction with WIPI3. Does knockdown of WIPI3 change FIP200 localization, association with ULK1, and affect FIP200 function in autophagy?*

- The reviewer correctly points out some of the many unanswered questions with regard to the function of FIP200 in the process of autophagy, e.g. it is unknown why FIP200 associates with TSC1 and why it is found in close proximity to lysosomes.
- We cannot answer all of these far-reaching questions in the current manuscript, but we provide data, that WIPI3 is found in complex with FIP200 in the LAMP2 compartment and at the nascent autophagosome, consistent with the reported localisations of FIP200 (albeit with currently non-identified functional consequences).
- We anticipate, however, that this notion will be helpful (for us and others) to further decipher the diverse roles of FIP200 in the process of autophagy.

5. *The authors identified critical residues in WIPI4 that mediate its ability to bind efficiently to ATG2A (N15A and D17A) and knockdown of these two gene causes increase in the size and number of puncta, so they conclude WIPI4-ATG2 complex regulates the size of autophagosomes. It would be very interesting to employ the non-binding mutants and look at the effects on autophagosome size. How does this complex control size of autophagosomes? Also of note, these residues are completely conserved across the WIPI proteins, so why the specificity for WIPI4?*

- The reviewer raised an important point with regard to the conservation of particular residues in WIPI proteins that confer differential associations, such as shown for WIPI2 specifically binding the ATG16L complex (Dooley et al., Mol Cell 2014).

- Here, we find a similar scenario for WIPI4 as both N15 and D17 (in WIPI4) are conserved throughout the WIPI members.
- We do not have an explanation for this, but we anticipate that further binding partners may contribute to such specificity.
- With regard to the further point on the WIPI4 mutant that does not bind ATG2 we can provide the following assessment (see below) that we conducted along with the results shown in the new Fig. 8g.
- In Fig. 8g we analysed a new WIPI4 mutant (D113A) that shows an impaired association with AMPK and in line with this, does not respond to glucose starvation with an increase in puncta formation. The corresponding text in the revised manuscript reads as follows:

"Further, using our panel of WIPI4 mutants (Suppl. Fig. 7a) we found that WIPI4 D113A was impaired in binding to AMPK (Suppl. Fig. 8c). Employing the GFP-WIPI4 D113A mutant (Fig. 8g, h), we found that this mutant did not respond to glucose starvation with an increase in the formation of punctate structures when compared to wild-type GFP-WIPI4 (Fig. 8g). Moreover, whereas GFP-WIPI4-puncta-positive structures significantly decreased in AMPK KD conditions (Fig. 8c, 8h), GFP-WIPI4 D113A punctate structures did not (Fig. 8h)."

- In this experiment (Fig. 8g) we also assessed the WIPI4 N15A mutant (data not shown) and found that this mutant also does not respond to glucose starvation by forming more punctate structures:

This result implies that both, AMPK and ATG2 prevents WIPI4 from localising at the nascent autophagosome. However, we did not include this data in the revised manuscript as we wish to address this follow-up in more detail in the future.

6. The authors demonstrate that endogenous AMPK and ULK1 associate with WIPI4-ATG2A more in fed conditions than nutrient or glucose starvation. What mediates this change in interaction? Does the LKB1 phosphorylation site on AMPK influence complex formation ie AMPK activation? Does kinase dead AMPK associate with the complex equally in starved conditions or alternatively no longer associate with WIPI4 in fed conditions? Does kinase dead ULK1 associate with WIPI4? Can either of these kinases phosphorylate WIPI4 or ATG2A?

- We greatly appreciate the reviewers' questions on this important topic. Indeed, these are intended studies in the laboratory.
- Towards this aim, we started to conduct SILAC-based phospho-proteomics comparing settings with and without AMPK. We identified several known and new phosphorylation sites on ATG proteins, amongst those an interesting phosphorylation site in ATG2 (S1453). Mutating this site in ATG2, however, showed no difference in the association with WIPI4 and AMPK (see below). The further analysis is ongoing and we very much hope that the reviewer may acknowledge that further assessments are part of subsequent investigations that include a variety of both ULK1 and AMPK variants such as kinase-dead mutants.

- We also found that WIPI4 can associate with ATG2 in the absence of AMPK, and that in fact very little AMPK phosphorylated by LKB1 is found in the WIPI4-ATG2 complex (data not shown). This observation underlines our suggestion that upon LKB1-mediated AMPK activation, WIPI4-ATG2 dissociates and translocates to the nascent autophagosome. Together with the characterization of our preliminary SILAC-based phospho-proteome analysis we will further dissect this setting in more detail in follow-up assessments.

7. How does binding to or release from AMPK-ULK1 modulate autophagosome size? Can authors speculate on this in discussion at least?

- We really thank the reviewer for suggesting to speculate on this point in the discussion. Accordingly we added our speculation in the discussion as follows:

“Upon starvation and AMPK activation, WIPI4-ATG2 dissociates from AMPK and ULK1 and localises at nascent autophagosomes, **potentially supporting further autophagosome maturation.**”

8. The authors performed a shRNA kinase library screen to identify kinases that effect WIPI1 puncta formation in starved conditions with a secondary screen of amino acid starvation. BRSK and NUAK were identified (although barely significant p values) and knockdown of these and AMPK decrease WIPI4 puncta slightly so they suggest AMPKRs regulate WIPI4. First, it is interesting that ULK1 was not pulled out of this screen, as it is a known regulator of autophagy that they showed can bind WIPI4. Do NUAK or BRSK bind to WIPI4? Does knockdown of NUAK or BRSK phenocopy WIPI4 knockdown at all?

- We deeply appreciate the reviewers' insightful comment. The kinome screening was conducted by omitting serum from full medium, and in this setting we have not identified ULK1. However, in our subsequent although preliminary kinome screens (that are not part of this study here) we used several starvation conditions and in such settings, indeed identified ULK1.
- However, in this screen here we identified expected kinases shown to influence autophagy, such as PDGFRB (Lei et al., Mol Cell Biol 2015).
- We also observed that in stable shRNA cell lines (secondary screen) where we downregulated kinases that we identified and selected in the primary screen (Suppl. Fig. 8a), effects on the number of WIPI1 puncta (our screening read-out) were not as prominent when compared to the results of the primary screen. Hence the resulting p-values in the secondary screen, e.g. in the case for PRKAG1 (AMPK γ), do not fully represent the significance in WIPI4 puncta reduction shown by transient siRNA-mediated downregulation of AMPK γ (Fig. 8c). Further work will be required for clarification of this observation.
- Indeed, downregulation of NUAK2 and BRSK2 phenocopies results that we achieved by assessing WIPI1, WIPI2 and LC3 puncta formation (data not shown). However, this analysis, along with the further characterization of the role of NUAK2 and BRSK2 in autophagy, is part of a subsequent study in our laboratory.

9. Some of the data presented (Supplementary Figure 4 on NudC and Figure 4 on WIPI1 and 2) could be excluded for the sake of clarity of the story. Both figures speak to potential interactors of WIP proteins in the autophagy pathway, but with the focus of the paper primarily on WIP3 and 4 interactions with TORC1 and AMPK pathways (as indicated by the title and subsequent figures) these data are unnecessary. The NudC data is the more interesting and novel of these 2 pieces of data, so would be the more interesting one to keep in the main paper if the paper is framed more as a full WIPI-autophagy-interactome paper. A section of supplementary information is dedicated specifically to go through detailed analysis of this figure, which seems like it should either be included as its own section in the manuscript or be omitted. If it is included some additional questions need commenting on by authors if not experimentally addressed:

A) Is NudC in a complex with all three WIPI (1,2,4) together or are their separate subsets of interactions that could potentially have different functions?

B) Do these proteins associate through their beta-propellers?

C) It would be nice to have quantitation of the increase in autophagosomes upon NudC knockdown.

D) If this complex negatively regulates autophagy, then one could imagine that NudC should dissociate from the WIPs under starvation conditions to relieve this repression, but that is not the case. How then do they become active (new binding partners, changes in localization perhaps?).

- We are indeed most grateful to the reviewers' suggestion to remove the NudC data from this manuscript, which we followed accordingly as this was indeed a minor point in the original presentation. We will assess this interesting new interaction in more detail in subsequent analysis. We have kept the confirmation of the association between WIPI1,2,4 and NudC but not between WIPI3 and NudC in the new Suppl. Fig. 4.

Minor points

1. Figure 1 title should include "under starvation"

- We thank the reviewer very much indeed for this suggestion and we discussed it in great detail. We felt that adding the term starvation may entirely reflect some of the new findings, as e.g. LC3-II was significantly accumulating in particular in fed conditions (new Fig. 1f). In addition, we found WIPI members differentially respond to glucose starvation (Fig. 8). From this we would like to keep the current title, however, including the alteration suggested by reviewer #2. Thus, we suggest the new title to read "WIPI3 and WIPI4 β -propellers act as scaffolds for LKB1-AMPK and the TSC complex in the control of autophagy".

2. Figure 1b left and right panels should be separated since they are different experiments in different cells, which is a bit confusing. Also would be worth swapping the data using endogenous WIPs from G361 cells to supplement and move all the overexpressed U2OS data to the main figure so 1B doesn't contain two different cell lines. Ideally it would be nice to western data to look at the relative expression of the endogenous proteins in G361 and U2OS cells the supplement as well.

- We very much agree with the reviewer that the display in the former Fig. 1b was not very clear. Accordingly, we separated the data on phospholipid binding and puncta formation to more clearly display the different experiments: phospholipid assessments are now all combined on the left, and the still images from our movies (Suppl. Movies 1-4) are now displayed in the right panels. We have, however kept the data achieved by visualizing endogenous WIPI1,2,4 in the main figure as we found that GFP-WIPI4 also marks binding to further phospholipids (see Suppl. Fig. 2d). In subsequent

studies we aim to assess the functional implications of WIPI4 to further phospholipids.

3. Consistent labeling of WIPI2B and WIPI2D throughout the figures would be helpful (instead of sometimes labeling just WIPI2).

- We thank the reviewer very much for this vital remark. We added the labelling that refers to the WIPI2 splice variants, WIPI2B and WIPI2D whenever possible. When using WIPI2 antibodies for example, one detects several WIPI2 isoforms and therefore at such settings we kept the labelling “WIPI2”.

4. For Figure 2b, it would be helpful to have quantitation of number of normal and abnormal autophagosomes in the different lines.

- We absolutely agree with the reviewer on this important point, that was also raised by reviewer #1 (please see above). We have readdressed the EM data and now provide a quantification (Suppl. Table 1, excel sheet labelled “Suppl. Fig. 2”) as well as representative images of this quantification in the new Suppl. Fig. 2b-f. These data more clearly demonstrate the different structures appearing in KD of the different WIPI proteins.
- The corresponding new text reads as follows (changes with regard to the original manuscript are highlighted in yellow):
“Compared with the control cell line (shControl, Fig. 2b, Suppl. Fig. 2b), WIPI1 KD cells did produce autophagosomes upon starvation (Fig. 2b, panel shWIPI1; Suppl. Fig. 2c), although we observed a 2.65-fold reduced presence of autophagosomal structures (Suppl. Table 1). In line with this finding, autophagic flux⁵⁰ assessments by LC3 lipidation analysis was previously reported to be negatively affected in WIPI1 KD cells²⁴. Additionally, we here found that starvation-induced degradation of long-lived proteins⁵² was significantly reduced in WIPI1 KD cells (Suppl. Fig. 2g). EM of WIPI2 KD revealed that proper autophagosome formation was negatively affected as inferred from a 4.93-fold accumulation of rough endoplasmic reticulum (RER) tubular structures (Fig. 2b, panel shWIPI2; Suppl. Fig. 2d), prominently marking an early blockade of autophagosome formation²⁶ prior to phagophore formation (template membranes from which autophagosomes emerge^{23, 53}) (Suppl. Table 1). In line with this finding, the appearance of autophagosomal structures decreased by 2.94 fold when compared to control cells (Suppl. Table 1). Moreover, in WIPI2 KD cells, the number of GFP-LC3 puncta was significantly reduced (Fig. 2c), corroborating the previous finding that WIPI2 is required for LC3 lipidation^{25, 26}. The above results confirm that both WIPI1 and WIPI2 function upstream of LC3, as previously suggested³⁵. In addition, the results underline the conception that WIPI2 is required for autophagosome formation, whereas WIPI1 is dispensable to a certain degree.
- Interestingly, both WIPI3 KD (Fig. 2b, panel shWIPI3; Suppl. Fig. 2e) and WIPI4 KD (Fig. 2b, panel shWIPI4; Suppl. Fig. 2f) resulted in the appearance of cup-shaped double-membrane structures resembling elongated phagophore formation sites (Fig. 2b, Suppl. Fig. 2e, f; Suppl. Table 1). In both, WIPI3 KD and WIPI4 KD cells the formation of autophagosomal structures decreased respectively by 6.42 and 3.47 fold when compared with shControl cells (Suppl. Table 1). In addition, RER tubular structures accumulated in the absence of WIPI4, as was observed in WIPI2 KD cells (Suppl. Fig. 2f, Suppl. Table 1).

5. Supplementary figure 2d, the way the Western is cropped, it looks like a shift of LC3 type but it is actually accumulation of the lipidated LC3 as stated in the text.

- We thank the reviewer very much for pointing this out. According to the request by reviewer #1 we have removed the LC3 lipidation data in WIPI1 KD and WIPI2 KD settings as this information was already reported previously (see above).

6. In Figure 5f, it would be helpful to have quantification of colocalization and to see if there is colocalization in fed cells as well.

- We very much thank the reviewer for this vital remark. We have discussed this issue above in our response to reviewer #1 and are here referring to page 6 in our point-to-point response.

7. Figure 5l, it is obvious that the re-feed is not activating TORC1 very well. Western is not very convincing

- We appreciate this critical remark. We altered the figure to highlight more the difference in TORC1-dependent ULK1 phosphorylation on S757 (see new

display of Fig. 5j). In fact, by using a variety of phospho-mTOR antibodies we observed that changes in mTOR autophosphorylation are not very prominent in different setting, in contrast to obvious differences in ULK1 phosphorylation. However, we kept the panel on mTOR autophosphorylation here as there is only a minor difference in starved conditions.

- We observe more mTOR at the lysosome in refed conditions when compared to starved conditions (Fig. 5k), however, also in starved conditions some mTOR can be detected to colocalise with LAMP2. This observation is in agreement with the new report on the extremely transient translocation of mTOR to the lysosomal surface by the Ktistakis laboratory (Manifa et al., eLIFE 2016).

8. *In Figure 8b, why is the knockdown assessed in the WIPI1-expressing U2OS line when the functional experiment (8c) is in the WIPI4 line (or is this a typo)?*

- We used this assessment in an exemplary fashion as we initially already addressed the functional assessment in all stable WIPI and LC3 U2OS cell lines. Using U2OS cell lines our comprehensive assessments of downregulation showed no differences in efficiencies between the parental U2OS cell line and all established, low expressing GFP-tagged variants.

9. *In multiple places in the paper, it is mentioned that WIPI3 and 4 have functions downstream of LC3 but in the model (8g) all the roles are depicted upstream or converging on LC3.*

- We absolutely agree with the reviewers' important point and have accordingly adjusted the model in the revised version of our manuscript.

Reviewer #4 (Remarks to the Author):

In this report by Proikas-Cezanne and co-workers, the authors describe a scaffolding function for WIPI3 and WIPI4 proteins to link LKB1-AMPK and TSC1-TSC2 signaling in the control of autophagy.

This is a well-conceived and well-designed study that leverages a kinome-wide screen with imaging and proteomics to identify factors involved in autophagosome formation. The functional proteomics data in the manuscript appear to be of high quality, and the mass spectrometry methods and approaches are well described.

We highly appreciate the reviewers' positive evaluation of our work and thank him/her very much indeed for the kind words and truly stimulating assessment.

Some additional discussion or experimentation would be helpful to further clarify or substantiate the claims of distinct protein complexes and linkages between protein interactors, rather than just relying on peptide counting, for example, "as we found only a few peptide counts for ATG16L in our WIPI1 MS analysis, we suggest that the interaction between WIPI1 and ATG16L occurs indirectly via WIPI2." It would be more convincing to perhaps knock down WIPI2 and demonstrate loss of ATG16L interaction by either mass spectrometry or western blotting.

- We wish to thank the reviewer very much for assessing our data on the interaction between WIPI2 and the ATG16L complex, including our suggestion that WIPI1 may support the WIPI2-dependent recruitment of the ATG16L complex (as elegantly shown by the Tooze laboratory; Dooley et al., Mol Cell 2014). We wish to address this issue in the following reasoning.

- We here suggest that WIPI1 supports WIPI2 in recruiting the ATG16L complex, since significantly less endogenous WIPI2-mediated ATG16L1 puncta are formed in WIPI1 KD settings (Fig. 4c).
- From our collective data we predict that WIPI2 is found prior to WIPI1 at the nascent autophagosome, and that this is required for subsequent WIPI1 binding, as less WIPI1 puncta are found in WIPI2 KD settings (Fig. 4b), while no differences in WIPI2 puncta formation was observed in WIPI1 KD settings (data not shown).
- In order to more clearly demonstrate the minor co-immunoprecipitation of ATG16L with WIPI1 we provide a new experiment in Fig. 4a that more strongly demonstrates that WIPI2B and WIPI2D prominently pull-down ATG16L, in contrast to WIPI1. This finding is in agreement with the data presented in Dooley et al (Dooley et al., Mol Cell 2014).
- We will, in subsequent investigations, investigate the possibility that WIPI1 supports WIPI2 by heterodimerization. For this, we will use WIPI1 and WIPI2 mutants to identify required amino acids that might confer heterodimerisation. We anticipate that such follow-up studies may shed further light on possible differences in the recruitment of WIPI2 and WIPI1 to the nascent phagophore, and the subsequent association between WIPI2 and the ATG16L complex.

Otherwise, I have no reservations about the quality of the mass spectrometry-based data in the manuscript.

- Once more we would like to thank the reviewer very much for assessing the quality of our proteome analysis.

Reviewers' Comments:

Reviewer #1 (Remarks to the Author)

The authors have addressed all my concerns.

Reviewer #2 (Remarks to the Author)

The authors did a great job in adequately addressing all my concerns and I am more than happy to recommend this manuscript for publication.

Reviewer #3 (Remarks to the Author)

This revised manuscript sufficiently addresses this reviewer's major concerns. Significant revisions and additions to the original manuscript help clarify the authors findings.

Reviewer #4 (Remarks to the Author)

Any concerns I had in prior reviews have been thoroughly addressed in revision.

NCOMMS-16-15136-T

Point-to-point response to the reviews on the manuscript

REVIEWERS' COMMENTS:

Reviewer #1 (Remarks to the Author):

The authors have addressed all my concerns.

Reviewer #2 (Remarks to the Author):

The authors did a great job in adequately addressing all my concerns and I am more than happy to recommend this manuscript for publication.

Reviewer #3 (Remarks to the Author):

This revised manuscript sufficiently addresses this reviewer's major concerns. Significant revisions and additions to the original manuscript help clarify the authors findings.

Reviewer #4 (Remarks to the Author):

Any concerns I had in prior reviews have been thoroughly addressed in revision.

We are most grateful to all four reviewers for their exceptionally helpful and supportive guidance throughout the review process, which enabled us to revise and significantly improve our manuscript. We thank the reviewer very much indeed for their final positive assessment of the revised version of our manuscript.